# TEST-TIME UNCERTAINTY ESTIMATION FOR MEDICAL IMAGE REGISTRATION VIA TRANSFORMATION EQUIVARIANCE

## ABSTRACT

Accurate image registration is essential for downstream applications, yet current deep registration networks provide limited indications of whether and when their predictions are reliable. Existing uncertainty estimation strategies, such as Bayesian methods, ensembles, or MC dropout, require architectural changes or retraining, limiting their applicability to pretrained registration networks. Instead, we propose a test-time uncertainty estimation framework that is compatible with any pretrained networks. Our framework is grounded in the transformation equivariance property of registration, which states that the true mapping between two images should remain consistent under spatial perturbations of the input. By analyzing the variance of network predictions under such perturbations, we derive a theoretical decomposition of perturbation-based uncertainty in registration. This decomposition separates into two terms: (i) an *intrinsic spread*, reflecting epistemic noise, and (ii) a *bias jitter*, capturing how systematic error drifts under perturbations. Across four anatomical structures (brain, cardiac, abdominal, and lung) and multiple registration models (uniGradICON, SynthMorph, TransMorph), the uncertainty maps correlate consistently with registration errors and highlight regions requiring caution. Our framework turns any pretrained registration network into a risk-aware tool at test time, placing medical image registration one step closer to safe deployment in clinical and large-scale research settings.

## 1 INTRODUCTION

Accurate alignment of images (image registration) is a cornerstone of contemporary medical imaging pipelines. It enables change detection between longitudinal scans for disease monitoring (Leung et al., 2010; Tustison et al., 2019), links patient anatomy to population atlases (Dubost et al., 2020), tracks lesions across longitudinal scans (Rokuss et al., 2025), propagates dose maps for radiation (Kessler, 2006), and underpins nearly every quantitative morphometric analysis. Modern learning-based approaches can often achieve real-time registration with performance comparable to, and in some benchmark settings exceeding, that of classical methods (Hering et al., 2022; Chen et al., 2025). However, most of them remain *opaque*: they predict a deformation field from an image pair, but *not an estimate of how reliable that field is*. Visual quality control of registrations is extremely labor intensive (particularly in 3D) and sometimes unfeasible (e.g., in real-time applications). Therefore, silent misregistrations can corrupt downstream measurements and go unnoticed until late in clinical or research workflows.

In safety-critical settings, identifying regions where predictions are unreliable is as important as producing the registration itself. To this end, uncertainty estimation in medical image registration has traditionally relied on probabilistic models, such as Bayesian inference or bootstrap sampling, to approximate posterior distributions over deformation parameters (Simpson et al., 2013; Kybic, 2009; Le Folgoc et al., 2017; Risholm et al., 2013; Wassermann et al., 2014; Gal & Ghahramani, 2015; Lakshminarayanan et al., 2017). While effective, these approaches are computationally expensive and do not scale well. In the deep learning era, most methods adapt classical ideas such as variational inference or ensembles (Dalca et al., 2019; Sedghi et al., 2019; Yang et al., 2017; Gong et al., 2022; Smolders et al., 2022), but they typically underestimate uncertainty, fail to capture the spatial structure of deformation fields, and remain tightly coupled to specific architectures. Moreover, these approaches often require retraining (sometimes with access to the original training data)

Figure 1: We present a test-time, model-agnostic framework for uncertainty estimation in image registration. The method leverages the principle of transformation equivariance: while true transformations remain equivariant under input perturbations, the learned registration networks often violate this property due to prediction errors. Concretely, we *(i)* perturb the source image $I^{A'} = I^A \circ \tau$, *(ii)* compose the predicted transformations into a common space $\tau \circ \hat{\phi}^{A'B}$, and *(iii)* measure their variance $Cov_\tau[\tau \circ \hat{\phi}^{A'B}]$ to produce spatially resolved uncertainty maps. The variance naturally decomposes into two interpretable components: Intrinsic Spread, reflecting inherent prediction variability, and Bias Jitter, capturing shifts in the estimated transformations under perturbations.

or architectural modifications, making them impractical for pretrained registration models. More recent voxel-level and hierarchical formulations (Zhang et al., 2024; Gopinath et al., 2024; Hu et al., 2025) improve expressiveness but still demand specialized training and are likewise tied to specific architectures. Moreover, little is known about how estimated uncertainties correlate with actual registration errors (Luo et al., 2019; 2020). As a result, most deployed registration networks operate as 'black boxes', offering little indication of when or where their outputs may fail, or providing such indication only at a high cost, often requiring retraining of the model or architectural modifications.

**Our perspective.** To address these limitations, we propose a test-time, model-agnostic uncertainty estimator (Fig. 1) with a principled decomposition that applies directly to pretrained networks (e.g., uniGradICON Tian et al. (2024b), SynthMorph Hoffmann et al. (2021)) without retraining, yielding interpretable and practically relevant uncertainty maps. Specifically, the proposed simple test-time uncertainty estimation framework is grounded in the *transformation equivariance* property of registration (Fig. 2), stating that the true mapping between image pairs should compose consistently with perturbations of the input. By analyzing the variance of predictions under such perturbations, we derive a principled estimator that decomposes into two terms: (*i*) an *intrinsic spread* reflecting epistemic noise (**blue arrow**→; Fig. 1), and (*ii*) a *bias jitter* term capturing systematic error variance under perturbations (**purple arrow**→ ; Fig. 1). Crucially, the framework requires **no retraining or architectural changes**, making it immediately applicable to pretrained registration models, even those that are not equipped with probabilistic formulations.

In summary, our main contributions are:

- We introduce a test-time uncertainty estimation framework that can be applied to any registration network without retraining or model modification (Section 4).
- We present the first theoretical analysis of the test-time perturbation-based uncertainty in image registration, decomposing the perturbation-induced variance with respect to registration error and revealing the relationship between the two. (Section 3).
- We conduct extensive experiments across four anatomical structures (brain, cardiac, abdominal, and lung) and three registration backbones (uniGradICON, SynthMorph, TransMorph) to evaluate (*i*) the correlation between the estimated uncertainty maps and the true registration error (Section 5.1 and 5.2), (*ii*) comparisons between the proposed method and one existing uncertainty estimation approach (Section 5.3), and (*iii*) two case studies with real transformations and anatomically inconsistent image pairs (Section 5.4).

Together, these results suggest that perturbation-based uncertainty estimation provides a simple, model-agnostic tool to probe the reliability of registration networks at test time, moving dense registration one step closer to safe deployment in clinical and large-scale research settings.

## 2 RELATED WORK

**Learning-based medical image registration.** Medical image registration aims to estimate a spatial transformation that aligns two given images. Classical registration methods (Avants et al., 2008; Klein et al., 2009; Modat et al., 2010; Heinrich et al., 2012) typically formulate this as an opti-

mization problem over a set of parameters describing the transformation. They can be used for a wide variety of registration tasks and can be highly accurate, but are often slow as they estimate registration parameters from scratch for every registration pair by numerical optimization. More recent supervised (Yang et al., 2017; Cao et al., 2017; Sokooti et al., 2017) and unsupervised (De Vos et al., 2017; Balakrishnan et al., 2019) learning-based registration approaches *predict* spatial correspondences much faster using a deep registration network. These learning-based approaches have achieved significant accuracy improvements by advanced transformation models (Shen et al., 2019; Niethammer et al., 2019; Tian et al., 2024a), network architectures (Mok & Chung, 2020a; Chen et al., 2022), training schemes (Hering et al., 2019; De Vos et al., 2019; Shen et al., 2019; Mok & Chung, 2020b), similarity measures (Tian et al., 2023b; Mok et al., 2024; Song et al., 2024), and regularization (Greer et al., 2021; Tian et al., 2023a). However, compared to the volume of work on learning-based registration networks, few works have explored uncertainty estimation in medical image registration, as we discuss in the following paragraphs. Moreover, the diversity in architectures (e.g., multi-resolution (Mok & Chung, 2020b) or multi-step (Shen et al., 2019) networks), the difficulty in balancing the similarity measure and the regularization loss (Tian et al., 2023a), and the difficulty to generalize to various tasks (Tian et al., 2024b) make it nontrivial to combine existing uncertainty estimation methods with registration networks. This motivates us to explore an uncertainty estimation approach for pre-trained registration networks.

**Uncertainty estimation in deep learning.** Uncertainty estimation aims to estimate how confident a neural network is in the prediction, serving as an important tool when deploying networks to the real world. This is crucial in fields that have strong requirements for reliability and robustness, such as medical imaging. Classical Bayesian approaches approximate the posterior distribution of model parameters via variational inference or stochastic regularization such as MC-dropout (Gal & Ghahramani, 2015). Deep ensembles provide another powerful non-Bayesian alternative (Lakshminarayanan et al., 2017; Rupprecht et al., 2017). Consistency (Moon et al., 2020; Li et al., 2023) has also been used to measure uncertainty. Other works address calibration and aleatoric versus epistemic uncertainty (Kendall & Gal, 2017). These techniques offer valuable insights but typically require architectural changes, retraining (and possibly access to the original data), or significant computational cost, making them difficult to deploy on pretrained models. Different from the Bayesian approaches, an alternative approach estimates uncertainty by measuring output variance across repeated forward passes under test-time perturbations (Ayhan & Berens, 2018; Wang et al., 2019). These test-time augmentation (TTA) strategies are attractive because they are model-agnostic and require no retraining. However, prior applications have largely been restricted to classification and segmentation, where the theoretical meaning of perturbation-induced variance remains unclear. Our work connects this paradigm to registration-specific geometric characteristics, which are transformation-equivariance.

**Uncertainty estimation for medical image registration.** Uncertainty estimation for image registration has long been an object of study. In the classical literature, probabilistic modeling approaches estimate uncertainty by computing (exactly or approximately) posterior distributions over deformation parameters via Bayesian inference (Simpson et al., 2013; Kybic, 2009; Le Folgoc et al., 2017; Risholm et al., 2013; Le Folgoc et al., 2016; Agn & Van Leemput, 2019) or Monte Carlo sampling (Iglesias et al., 2013). Alternatively, bootstrap sampling has been employed as an empirical ensemble strategy to quantify uncertainty (Kybic, 2009).

In the deep learning era, however, uncertainty estimation remains largely underexplored. Most approaches either directly adapt classical techniques, e.g., Bayesian inference or ensembles (Yang et al., 2017; Gong et al., 2022; Smolders et al., 2022; Chen et al., 2024a) (see also the survey by Chen et al. 2024b) or yield uncertainty as simplistic by-products, such as the variational inference strategy in Dalca et al. (2019); Sedghi et al. (2019), which is known to underestimate uncertainty. Since these methods often fail to capture the spatial distribution of deformation fields, more recent work has proposed aleatoric formulations tailored to registration (Zhang et al., 2024), but their scope remains limited to voxel-level noise modeling. Recent studies (Gopinath et al., 2024; Hu et al., 2025) have modeled uncertainty hierarchically, tracing its propagation across network outputs, transformation parameters, and downstream tasks. Such approaches enable sampling, exploration of modes of variation, and principled error bar estimation. While promising, these approaches typically require custom training protocols and remain tightly bound to specific architectures. Furthermore, they rarely provide theoretical clarity on what the predicted variance represents, leaving many uncertainty maps as task-specific heuristics with limited interpretability. Finally, another important gap

in existing methods is the limited investigation of how estimated uncertainties correlate with actual registration errors (Luo et al., 2019; 2020). This may be partly due to the scarcity of datasets with manual annotations of landmark pairs that could serve as a gold standard.

In contrast with these methods, we introduce a test-time, model-agnostic uncertainty estimation for pre-trained registration networks. This design not only facilitates broad interpretability but also ensures applicability to large registration models such as uniGradICON and SynthMorph, without the need for retraining.

## 3 THEORETICAL ANALYSIS OF UNCERTAINTY ESTIMATION VIA INPUT PERTURBATIONS

### 3.1 BACKGROUND

Let $f_\theta(I^A, I^B)$ denote a registration network that estimates the transformation $\hat{\phi}^{AB} : \Omega^B \to \Omega^A$ from the domain of the target image $I^B : \Omega^B \to \mathbb{R}$ to the source image $I^A : \Omega^A \to \mathbb{R}$. For a given target point $y \in \Omega^B$, the network predicts its corresponding location in the source space, *approximating* the true transformation $\phi^{AB}$,

$$\hat{\phi}^{AB}(y) := f_\theta(I^A, I^B)(y) = \phi^{AB}(y) + \epsilon(y), \tag{1}$$

where $\epsilon(y)$ represents the error of the network output. Note that we use $\hat{\phi}$ to represent the prediction of the registration network and $\phi$ to represent the true transformation.

**Transformation equivariance for image registration solver.** Due to the inherent properties of the registration task, the anatomical correspondence between $I^A$ and $I^B$ should remain unchanged to a spatial perturbation applied to $I^A$, whether rigid or diffeomorphic. In other words, the *true transformation* from $I^B$ to $I^A$ can be written as the composition of the perturbation $\tau : \Omega^{A'} \to \Omega^A$ with the true transformation from $I^B$ to the perturbed source $I^{A'} := I^A \circ \tau$ (as shown in Fig. 2). Formally, we expect

$$f(I^A, I^B) = \tau \circ f(I^A \circ \tau, I^B), \tag{2}$$

or equivalently,

$$\phi^{AB}(y) = \tau \circ \phi^{A'B}(y), \tag{3}$$

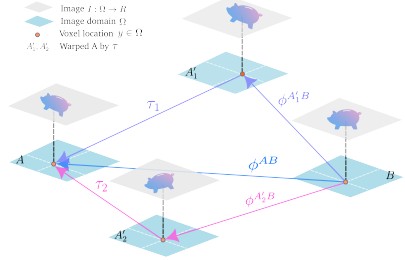

$$\phi^{AB}(y) = (\tau_1 \circ \phi^{A'_1 B})(y) = (\tau_2 \circ \phi^{A'_2 B})(y)$$

Figure 2: Transformation equivariance for image registration solver.

where $f(\cdot, \cdot)$ denotes the *ideal registration solver* that returns the true transformation $f(I^A, I^B) = \phi^{AB}$. Here, the ideal solver is an abstract construct and does not correspond to any particular algorithmic implementation (e.g., classical optimization or a learned registration network).

> The transformation equivariance (Eq. (2)) holds for the *ideal solver* $f(\cdot, \cdot)$ but is not necessary for a registration network $f_\theta(\cdot, \cdot)$. This discrepancy motivates our central question: **can the variance induced by perturbations $\tau$ be exploited to estimate uncertainty?**

### 3.2 INPUT PERTURBATION AND COMPOSITION FOR REGISTRATION NETWORKS $f_\theta(\cdot, \cdot)$

Now consider a transformation $\tau \in \text{Diff}(\Omega)$ and $\tau : \Omega^{A'} \to \Omega^A$. Applying the registration network to the perturbed image pair $(I^{A'}, I^B)$, we obtain

$$\hat{\phi}^{A'B}(y) := f_\theta(I^A \circ \tau, I^B)(y) = \phi^{A'B}(y) + \epsilon_\tau(y). \tag{4}$$

Given that $\tau$ is diffeomorphic, $\tau^{-1}$ exists. With Eq. (3), we have

$$\phi^{A'B}(y) = \tau^{-1} \circ \phi^{AB}(y), \tag{5}$$

substituting $\phi^{A'B}(y)$ in Eq. (4) with Eq. (5), we have

$$\hat{\phi}^{A'B}(y) := f_\theta(I^A \circ \tau, I^B)(y) = (\tau^{-1} \circ \phi^{AB})(y) + \epsilon_\tau(y). \tag{6}$$

By composing the perturbed transformation $\tau$ and the output $\hat{\phi}^{A'B}(y)$ of the registration network given the perturbed image pair, we have

$$g_\tau(y) := \tau \circ \hat{\phi}^{A'B}(y) = \tau((\tau^{-1} \circ \phi^{AB})(y) + \epsilon_\tau(y)). \tag{7}$$

Compared to Eq. (3), Eq. (7) has an extra term $\epsilon(y)$. In the following section, we analyze the relationship between the mean and variance of $g_\tau(y)$ and the error of the prediction $\epsilon(y)$.

### 3.3 Mean and Variance of $g_\tau(y)$

We now analyze the expected behavior of the composed outputs $g_\tau(y)$ across random perturbations $\tau \sim \mathcal{T}$, under the assumption that the registration error $\epsilon$ follows a distribution characterized by mean and covariance.

**Error model.** Given a registration output under a perturbed source input,

$$\hat{\phi}^{A'B}(y) = (\tau^{-1} \circ \phi^{AB})(y) + \epsilon_\tau(y), \tag{8}$$

we assume the residual error $\epsilon_\tau(y) \in \mathbb{R}^d$ follows a distribution $\mathcal{P}$ characterized by mean $\mu_\epsilon(\tau; y)$ and covariance $\Sigma_\epsilon(\tau; y)$[1],

$$\epsilon_\tau(y) \sim \mathcal{P}, \qquad \mu_\epsilon(\tau; y) \in \mathbb{R}^d, \ \Sigma_\epsilon(\tau; y) \in \mathbb{R}^{d \times d}. \tag{9}$$

We analyze the composed prediction $g_\tau(y)$ under three perturbation classes: *(i) general diffeomorphisms*, *(ii) affine maps*, and *(iii) translations*. Detailed proofs are provided in Appendix B.

**Lemma 3.1** (Mean and covariance under an arbitrary diffeomorphic perturbation)**.** *Let the perturbed output be* $g_\tau(y) = \tau\big((\tau^{-1} \circ \phi^{AB})(y) + \epsilon_\tau(y)\big)$, *with* $\epsilon_\tau(y) \sim \mathcal{N}\big(\mu_\epsilon(\tau; y), \Sigma_\epsilon(\tau; y)\big)$, *and denote by* $J_\tau(y) := D\tau_u\big|_{u = \phi^{AB}(y)} \in \mathbb{R}^{d \times d}$ *the Jacobian of the perturbation at voxel* $y$. *Without assuming any independence between* $\tau$ *and* $\epsilon_\tau$ *we have*

$$\mathbb{E}_\tau\big[g_\tau(y)\big] = \phi^{AB}(y) + \mathbb{E}_\tau\big[J_\tau(y)\,\mu_\epsilon(\tau; y)\big], \tag{10}$$

$$\mathrm{Cov}_\tau\big[g_\tau(y)\big] = \mathbb{E}_\tau\big[J_\tau(y)\,\Sigma_\epsilon(\tau; y)\,J_\tau^\top(y)\big] + \mathrm{Cov}_\tau\big[J_\tau(y)\,\mu_\epsilon(\tau; y)\big]. \tag{11}$$

**Interpretation.** The first term in Eq. (11) corresponds to the zero-mean epistemic (and possible aleatoric) variance intrinsic to the network (**blue arrow**; Fig. 1). The second term captures how the center of the error distribution shifts under input perturbations (**purple arrow**; Fig. 1). We refer to these components as *intrinsic spread* and *bias jitter*, respectively. The *intrinsic spread* reflects epistemic (and aleatoric) variability of the residual, linearly distorted by each sampled Jacobian and averaged over the perturbation distribution. In contrast, the *bias jitter* quantifies the covariance of the residual mean $\mu_\epsilon(\tau; y)$ when warped by different transformations.

**How does the error model affect the variance?** We model the registration error as a distribution without loss of generality in our analysis. For a deterministic registration network that outputs a single transformation, the error model falls back to a constant, and thus the residual variance vanishes ($\Sigma_\epsilon = 0$), so the perturbation covariance reduces to

$$\mathrm{Cov}_\tau\big[g_\tau(y)\big] = \mathrm{Var}_\tau\big[J_\tau(y)\,\mu_\epsilon(\tau; y)\big].$$

Thus, the same decomposition still applies, meaning the uncertainty map reflects perturbation-sensitive *bias jitter* even when intrinsic variance is absent.

**Lemma 3.2** (Mean and variance under affine perturbation)**.** *Let* $\tau(z) = Az + b$ *be a random affine transformation with* $A \in \mathbb{R}^{d \times d}$, $b \in \mathbb{R}^d$. *Since the translation part cancels,* $g_\tau(y) = \phi^{AB}(y) + A\epsilon_\tau(y)$. *Then:*

$$\mathbb{E}_{A,b}\big[g_{A,b}(y)\big] = \phi^{AB}(y) + \mathbb{E}_A\big[A\,\mu_\epsilon(A; y)\big], \tag{12}$$

$$\mathrm{Var}_{A,b}\big[g_{A,b}(y)\big] = \mathbb{E}_A\big[A\,\Sigma_\epsilon(A; y)\,A^\top\big] + \mathrm{Var}_A\big[A\,\mu_\epsilon(A; y)\big]. \tag{13}$$

**Corollary 3.3** (Translation perturbations)**.** *Translations are the special case of Lemma 3.2 with* $A = I$ *and* $b = t \sim \mathcal{D}_t$. *Then* $g_t(y) = \phi^{AB}(y) + \epsilon_t(y)$, *and*

$$\mathbb{E}_t[g_t(y)] = \phi^{AB}(y) + \mathbb{E}_t\big[\mu_\epsilon(t; y)\big], \tag{14}$$

$$\mathrm{Var}_t[g_t(y)] = \mathbb{E}_t\big[\Sigma_\epsilon(t; y)\big] + \mathrm{Var}_t\big[\mu_\epsilon(t; y)\big]. \tag{15}$$

**Interpretation.** In Eq. (13), the first term is the *intrinsic spread* propagated by $A$, and the second is the *bias jitter*, the covariance of residual means under different affines. When $A = I$, the affine perturbation reduces to a pure translation, with the Jacobian equal to the identity.

[1]We use $\mu_\epsilon(\tau; y)$ and $\Sigma_\epsilon(\tau; y)$ to denote the mean and covariance of the residual at voxel $y$ under perturbation $\tau$, noting they are equivalent to $\mu_\epsilon(y; \tau)$ and $\Sigma_\epsilon(y; \tau)$.

## 4 PERTURBATION-BASED UNCERTAINTY ESTIMATION

Building on the theory analysis in Section 3, we estimate uncertainty by applying random perturbations to the source image and measuring the variance of the composed predictions. The perturbation family can be chosen flexibly (e.g., translations, affine maps, elastic deformations), as is the registration network. Our framework is agnostic to these choices. The algorithm is described in Algorithm 1. The per-voxel uncertainty map is computed from the sample variance of the composed outputs. We report the scalar uncertainty $u(y) = \sqrt{\mathrm{tr}\, S(y)}$ (root trace), unless otherwise stated. Implementation details and statistics are provided in Appendix C.1.

---

**Algorithm 1:** Perturbation-based Registration Uncertainty

---

**Input:** Source $I^A$, target $I^B$, network $f_\theta$, samples $N$, perturbation distribution $\mathcal{P}(\tau)$
**Output:** Mean field $\mu(y)$, uncertainty map $u(y)$

1 **for** $n = 1$ **to** $N$ **do**
2     sample $\tau_n \sim \mathcal{P}(\tau)$
3     $I_n^{A'} \leftarrow \mathrm{SpatialTransform}(I^A, \tau)$
4     $\hat{\phi}_n^{A'B} \leftarrow f_\theta(I_n^{A'}, I^B)$
5     $g_n \leftarrow \mathrm{Compose}(\tau, \hat{\phi}_n^{A'B})$
6 $\mu(y) \leftarrow \frac{1}{N}\sum_n g_n(y)$
7 $S(y) \leftarrow \frac{1}{N}\sum_n \|g_n(y) - \mu(y)\|^2$
8 $u(y) \leftarrow \sqrt{\mathrm{tr}\, S(y)}$
9 **return** $\mu(y)$, $u(y)$

---

## 5 EXPERIMENTS

We conduct extensive experiments to evaluate the proposed test-time uncertainty estimation at both the dataset (Section 5.1) and single-case levels (Section 5.2). Each experiment examines the correlation between the predicted uncertainty maps and the true registration errors across varying ground-truth transformations (translation, affine, deformation), perturbation types (translation, scale, shear, deformation), anatomical structures (brain, abdomen, cardiac), and registration networks with different backbones. Then, we compare our proposed uncertainty estimation with the uncertainty estimated with MC-dropout (Section 5.3). Lastly, we qualitatively analyze two case studies (Section 5.4) where uncertainty maps highlight anatomically inconsistent or high-risk regions to justify our observations. Please find more details of metrics, datasets, and backbone networks in Appendix C.2.

**Datasets.** We use a brain MRI dataset curated from 11 public sources (see Appendix C.2 for details), the IXI brain MRI dataset (Chen et al., 2022; IXI), the ACDC cardiac MRI dataset (Bernard et al., 2018), and the Learn2Reg abdomen CT dataset (Xu et al., 2016) in the quantitative experiments. In addition, we conduct case studies on Brats-Reg (Baheti et al., 2021) and Learn2Reg ThoraxCBCT (Hugo et al., 2016; 2017) datasets.

**Backbones.** We conduct experiments on two pretrained deterministic foundation models chosen for their strong generalization and one task-specific probabilistic model. The first is uniGradICON (Tian et al., 2024b), a foundation registration model spanning multiple anatomies. The second is the nonrigid registration network from SynthMorph (Hoffmann et al., 2021), a contrast- and resolution-agnostic registration model for the brain. The third is the probabilistic (MC-dropout) registration model from TransMorph (Chen et al., 2022).

### 5.1 DATASET-LEVEL QUANTITATIVE ANALYSIS

To evaluate correlation with ground-truth error, we first simulate paired images using three categories of deformations: (i) translations, (ii) affine transformations combining translation, shear, and scale, and (iii) elastic B-spline deformations. While these simulated transformations provide controlled conditions, they do not fully capture the challenges of real registration. To approximate real scenarios while preserving access to ground truth, we also use ANTs (Avants et al., 2008) to estimate affine and nonrigid transformations between real image pairs; the estimated transformations are then applied to warp the source image to form new target images. This ensures that the ground-truth mappings are derived from real registrations while remaining accessible for evaluation. Both settings follow the same experimental design, applying perturbations of different types (translation, scale, shear, and deformation) to assess uniGradICON and SynthMorph. Details of the experiment settings are provided in Appendix C.3, and the results are shown in Fig. 3.

**Result and discussion.** Fig. 3 shows results for brain and abdominal datasets. For **uniGradI-CON**, the estimated uncertainty correlates positively with the registration error across all settings (Fig. 3a (a)-(h)). Correlations are strong when the ground truth is linear (translation or affine; Fig. 3a

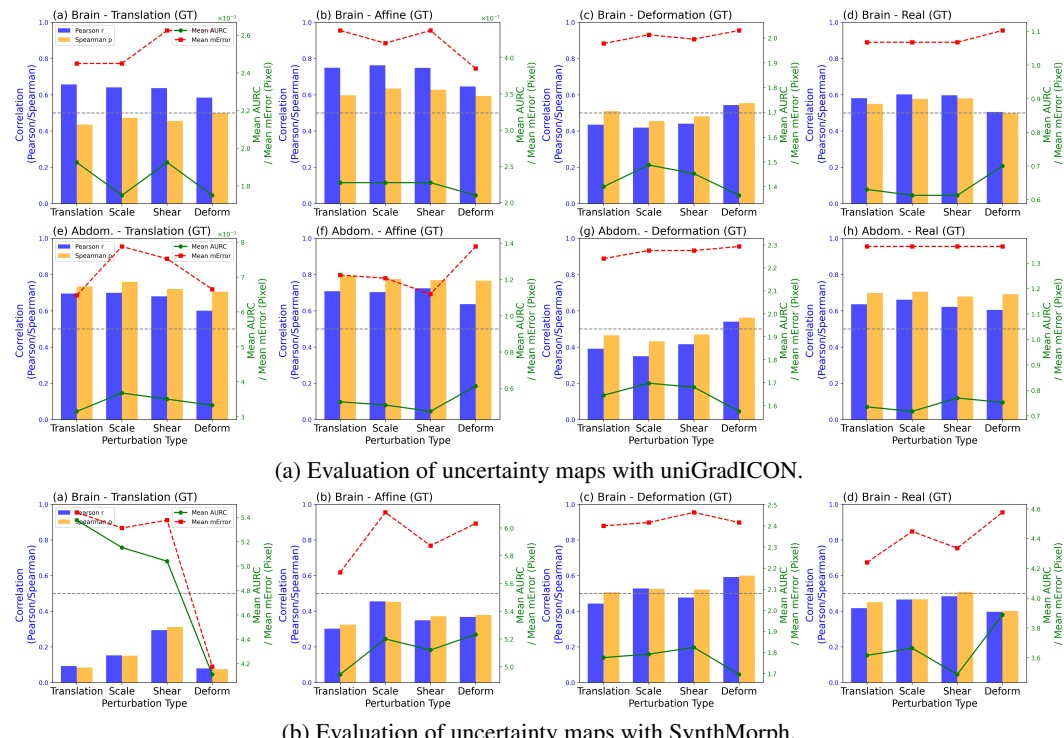

(a) Evaluation of uncertainty maps with uniGradICON.

(b) Evaluation of uncertainty maps with SynthMorph.

Figure 3: Correlation between the estimated uncertainty and the true registration error for a combination of varying ground truth transformation types (GT), perturbation transformation types, datasets, and registration model with different backbones.

(a,b,e,f)) and moderate when the ground truth is nonlinear (deformation or real; Fig. 3a (c,d,g,h)). This drop in correlation likely arises from two interacting factors: (i) nonlinear ground truths induce more localized errors, and (ii) linear perturbations (translation, scale, shear) are less effective at capturing such local discrepancies. See Section 5.2 for further analysis. Despite these differences, the AURC (green line) consistently improves over the mean error baseline (red line), confirming that the uncertainty map provides a more informative ranking of errors than chance. For **SynthMorph**, evaluated on brain MRI, strong correlations appear only under nonrigid ground truths (Fig. 3b (c)), while correlations under translation and affine dropped quickly (Fig. 3b (a,b)). This reflects the model's inductive bias: it is trained for nonlinear registration and not designed to handle large global motions, as also indicated by its higher error range compared to uniGradICON. When the task matches its capacity (Fig. 3b (c)), the correlations are consistent with uniGradICON. Under **real transformations**, both networks maintain moderate correlations with consistent AURC improvements (Fig. 3a (d,h); Fig. 3b (d)). The observed correlations fall between the linear (Fig. 3a (a,b,e,f) and Fig. 3b (a,b)) and nonlinear (Fig. 3a (c,g) and Fig. 3b (d)) ground truth transformation, which is as expected since real transformations combine global and local components. Moreover, the nonlinear part of real transformations is less extreme than the simulated deformations (Fig. 4a, last column), making them easier to capture. The informativeness of the uncertainty map depends on the network's inductive bias: foundation-style networks such as uniGradICON, which can model both linear and nonlinear transformations, perform well with all perturbation types, whereas SynthMorph, which specializes in nonlinear alignment, works better with deformation-based perturbations. This **limitation** is acceptable in practice, as the nonlinear SynthMorph network is not typically used in scenarios involving substantial linear transformations. **In sum**, these results demonstrate that perturbation-based uncertainty provides a reliable proxy for registration error. Please refer to Appendix C.3 for a more detailed discussion of why and when the uncertainty map correlates closely with the error map.

## 5.2 SINGLE-CASE QUANTITATIVE ANALYSIS

To further probe the behavior of the proposed uncertainty map, we conduct a single-case analysis using a randomly sampled subject from the brain MRI dataset, following the settings in Section 5.1.

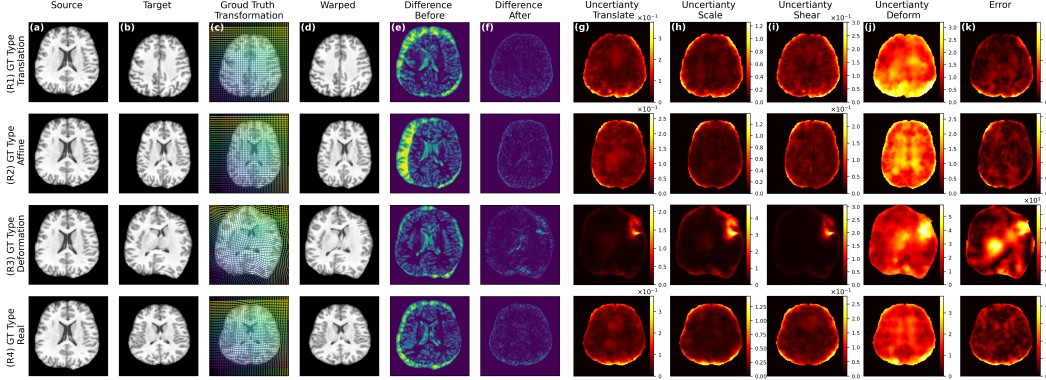

(a) The qualitative results for one randomly sampled brain MRI case.

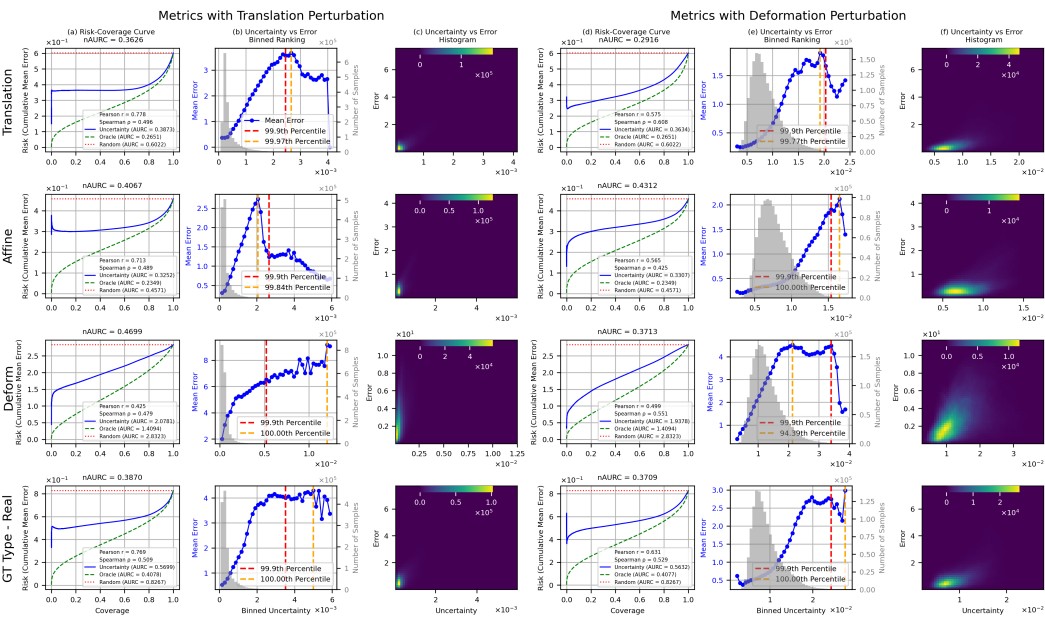

(b) The quantitative results for one randomly sampled brain MRI case.

Figure 4: The qualitative and quantitative results of the uncertainty map measured of a randomly sampled brain MRI image for a combination of varying ground truth transformation types (GT), perturbation transformation types with uniGradICON.

We evaluate the results qualitatively and quantitatively by visualizing uncertainty maps, error maps, and risk–coverage behavior, as well as reporting correlation and AURC metrics in Fig. 4.

**Result and discussion.** Fig. 4 shows qualitative and quantitative results for one randomly sampled brain MRI case. Under linear ground truths (translation and affine; Fig. 4a R1, R2), the uncertainty maps closely resemble the error maps ($k$-th column, denoted as C(k)), with the exception of the deformation-based uncertainty map (C(j)). In contrast, under a nonlinear ground truth (R3), the uncertainty map derived from nonlinear perturbations (C(j)) aligns more closely with the error than those derived from linear perturbations (C(g–i)), reflecting the interplay described in Section 5.1. Importantly, the error map under nonlinear ground truth (R3) exhibits more localized errors than those under linear ground truths (R1, R2). Nonlinear perturbations can reveal such local discrepancies, which translation-based perturbations fail to capture, underscoring their complementary strengths. The **quantitative results** over the full 3D volume (Fig. 4b) confirm these trends. Translation-based uncertainty achieves strong correlations with the true error ($r = 0.71$–$0.78$, $\rho = 0.49$–$0.50$) under translation, affine, and real ground truths, and moderate correlation ($r = 0.43$, $\rho = 0.48$) under deformation. To assess ranking ability, we compute normalized AURC, nAURC $= \frac{\text{Uncertainty}-\text{Oracle}}{\text{Random}-\text{Oracle}}$, which measures performance relative to random and oracle rankings. Lower values indicate better ranking. Across all ground-truth types, our method achieves low nAURC (0.29–0.47), demonstrating strong ranking capability. The second panel of each exper-

iment (Fig. 4b C(b,e)) visualizes the mean error across bins, sorted by uncertainty, with bin sizes indicated in gray. Most errors are correctly prioritized (94.4th to 99.9th percentile), supported by the histogram (Fig. 4b C(c,f)). Overall, the results demonstrate that the proposed confidence score reliably highlights misalignments, achieves a strong correlation under global transformations, maintains a reasonable correlation under nonlinear deformations, and consistently separates high- and low-error regions. Extended results on additional cases are provided in Appendix C.4.

### 5.3 COMPARISON WITH EXISTING UNCERTAINTY ESTIMATION

We used a deterministic model in the previous experiments because the two foundation models that support various experiment settings (e.g., different anatomical structures, ground truth translations) are deterministic, but our method and theory analysis are not limited to deterministic models. In this experiment, we evaluate how the proposed uncertainty performs with a probabilistic model. We use the publicly available TransMorph-Bayes model as the backbone, which is trained for atlas-MRI registration on IXI dataset. We use the IXI test set in this experiment. Since there is no groud truth for this dataset, we compute the correlation between the following three uncertainty maps: (1) MC-dropout uncertainty (MC Dropout); (2) Proposed Test-time uncertainty with dropout turned off (Our(static)); and (3) Proposed Test-time uncertainty with dropout on (Our(total)), and show the visualization of four examples in Fig. 5. The proposed uncertainty is highly correlated with the uncertainty of the MC Dropout approach, demonstrating the validity of our method relative to existing uncertainty estimation methods.

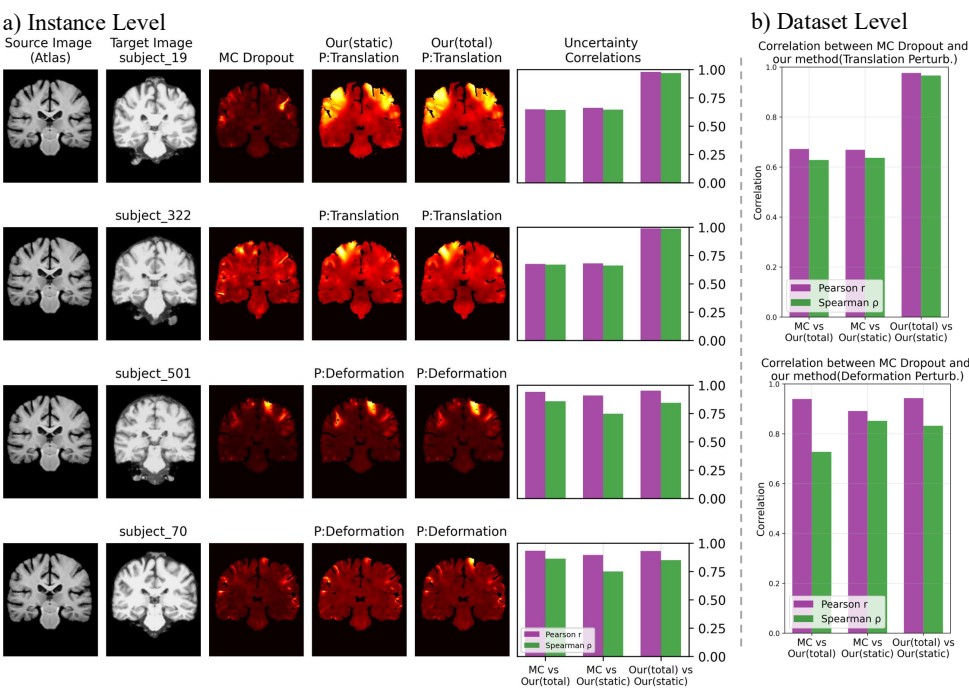

Figure 5: Comparison between our method and the MC-dropout.

### 5.4 CASE STUDY: UNCERTAINTY MAPS WITH ANATOMICAL INCONSISTENCIES

Beyond controlled perturbations, a key question is whether the proposed uncertainty map can highlight critical cases where anatomical correspondence is absent. We explore two case studies. We present one example for each case study and more examples are provided in Appendix C.5.

**Tumor resection.** Using one pre-/post-operative pair from BraTS-Reg (Baheti et al., 2021), we examine whether uncertainty reflects structural changes caused by surgery. Registrations and corresponding uncertainty maps are computed from T1-weighted images, while T2-weighted images are included to better delineate abnormal regions. Overlaying the uncertainty map on the target and warped T2 in Fig. 6a confirms that high-uncertainty regions align well with the location of the tumor.

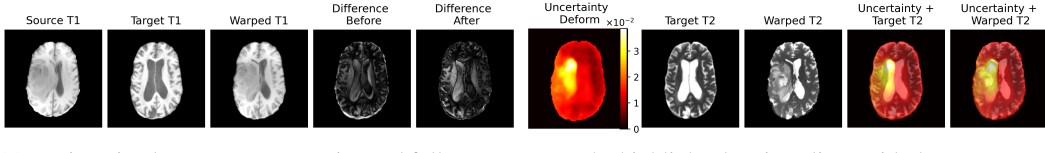

(a) Registration between pre-operative and follow-up MRIs. The highlighted region aligns with the tumor area.

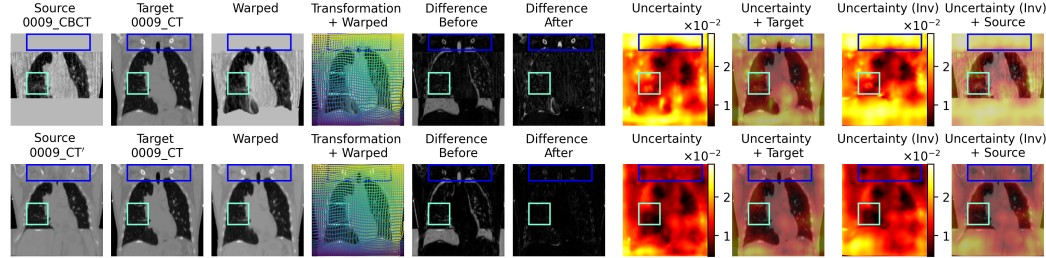

(b) Registration between CBCT and CT, which have different FOVs. Compared to the synthetic CT/CT control pair with matched anatomy, the CBCT/CT setting shows elevated uncertainty in regions outside the CBCT field of view (blue box) and in noisy regions characteristic of CBCT (light blue box). Uncertainty (Inv) denotes the uncertainty map warped via the inverse of $\phi^{AB}$.

Figure 6: Case studies of uncertainty maps used when there exist anatomical inconsistencies.

**Different fields of view (FOV).** We investigate whether the uncertainty map can highlight regions of missing correspondence caused by different FOVs across imaging modalities (e.g., CBCT and CT). We use data from the Learn2Reg ThoraxCBCT dataset (Hugo et al., 2016; 2017). Specifically, we register one CBCT/CT pair and compute the uncertainty map. To create a control comparison, we generate a synthetic source image $CT'$ by warping the CT with the transformation estimated by uniGradICON during CBCT/CT registration. We then register $CT'$ with the original CT, a setting in which no FOV-related inconsistencies are present. Compared to the synthetic CT/CT control pair with matched anatomy, the CBCT/CT setting shows elevated uncertainty in regions outside the CBCT FOV (blue box) and in noisy regions characteristic of CBCT (light blue box) in Fig. 6b.

Both case studies demonstrate that the proposed uncertainty map serves as a practical quality-control tool, as it not only correlates with registration error, but also flags clinically relevant inconsistencies (e.g., resection cavities, modality-induced gaps) that conventional similarity measures may overlook.

## 6 CONCLUSION

We introduced a test-time uncertainty estimation framework for image registration that leverages transformation equivariance, supported by both rigorous theoretical analysis and extensive empirical validation across four anatomical structures using two deterministic and one probabilistic registration model. In addition, our approach produces spatially resolved uncertainty maps that align with true registration errors and highlight anatomically inconsistent regions in the case study, all without retraining the models. Our study has several limitations. The current theory relies on first- and second-order moments of the error model, leaving higher-order characteristics such as skewness or kurtosis unexplored. Second, our method does not disentangle epistemic and aleatoric uncertainty in the manner of Bayesian approaches; a theoretical examination of our uncertainty map from a Bayesian perspective may offer deeper insight. Furthermore, the registration errors may be underestimated by the proposed method for certain types of transformation-equivariant registration networks (e.g., translation- or rotation-equivariant registration networks (Greer et al., 2025)). In such cases, perturbations that fall outside the supported equivariant transformations should be chosen. Lastly, our study focuses on pretrained registration networks, and we have not examined the effectiveness of the proposed method for conventional registration algorithms that rely on iterative optimization. Despite these caveats, our results consistently show that perturbation-based uncertainty provides a reliable and model-agnostic approach for registration reliability. By converting equivariance violations into actionable risk indicators, this framework opens the door to practical uncertainty-aware registration pipelines for clinical and scientific applications.

## 7 ETHICS AND REPRODUCIBILITY STATEMENT

### 7.1 ETHICS STATEMENT

This work focuses on developing a test-time uncertainty estimation framework for medical image registration. All experiments are conducted on publicly available, de-identified datasets. No new human or animal data were collected, and therefore no additional IRB approval was required.

The proposed methodology is intended as a research contribution toward improving the safety and reliability of registration in clinical and large-scale biomedical research settings. In particular, the uncertainty maps are designed to provide interpretable indicators of registration reliability, thereby reducing the risk of unintended misuse of misaligned images in downstream tasks. The method is not intended for direct clinical deployment without further validation in prospective studies.

### 7.2 REPRODUCIBILITY STATEMENT

A complete mathematical description of the proposed uncertainty estimation framework, including proofs of the theoretical results, is provided in the main text (Section 3) and in detail in the appendix (Appendix B). The datasets used in our experiments are all publicly available. The details of the datasets, evaluation metrics, and backbone registration networks are described in Section 5 and elaborated in Appendix C.2. The exact experimental settings, including parameter ranges for the perturbations, simulated ground truth deformations are documented in Appendix C.1 and Appendix C.3. We will release the implementation upon acceptance of this work.

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

ACKNOWLEDGMENTS

We use LLM to improve the writing quality and grammar check of the manuscript. The LLM is also used to generate Python matplotlib visualization code.

OVERVIEW OF THE APPENDIX

The **appendix** is organized into the following sections, each expanding on different aspects of the proposed transformation–perturbation-based uncertainty estimation:

- **Appendix A** explains all notations used throughout the paper.
- **Appendix B** presents detailed proofs for the mean and covariance under different types of transformation perturbations.
- **Appendix C.1** lists the details of the transformation perturbation and the statistics of our approach (e.g., runtime and memory consumption).
- **Appendix C.2** describes the metrics, datasets, and backbone registration networks employed in the experiments.
- **Appendix C.3** details the experimental setup used in Section 5.1, including parameter choices, metric definitions, and figure generation. This section also discusses why the uncertainty map correlates with the error map and extends the dataset-level experiments to additional anatomical structures (e.g., cardiac).
- **Appendix C.4** provides additional single-case analyses on randomly sampled brain, abdomen, and cardiac images to demonstrate the generalization of our observations.
- **Appendix C.5** gives an extended discussion of the experiments and results in Section 5.4, and includes further case studies on BraTS-Reg and Learn2Reg ThoraxCBCT to illustrate the generalizability of our findings.

## A    APPENDIX: NOTATIONS

| Symbol | Meaning |
|--------|---------|
| $I^A, I^B$ | Source and target images |
| $\Omega$ | Image domain |
| $\mathbb{I}$ | Identity transformation |
| $\tau \in \mathrm{Diff}(\Omega)$ | Random perturbation sampled from a diffeomorphic transformation family |
| $\phi^{AB}$ | True transformation mapping points from $\Omega^B$ to $\Omega^A$ |
| $\hat{\phi}^{AB}$ | Transformation predicted by the registration network $f_\theta(\cdot, \cdot)$ |
| $\epsilon(y)$ | Registration error at voxel $y$ |
| $\mu_\epsilon(\tau; y)$ | Mean of the error under perturbation $\tau$ |
| $\Sigma_\epsilon(\tau; y)$ | Covariance of the error under perturbation $\tau$ |
| $g_\tau(y)$ | Composed prediction under perturbation, $g_\tau(y) = \tau \circ \hat{\phi}^{A'B}(y)$ |
| $J_\tau(y)$ | Jacobian of the perturbation $\tau$ at location $y$ |
| $\mu(y)$ | Mean of composed outputs over perturbations |
| $S(y)$ | Sample covariance of composed outputs over perturbations |
| $u(y)$ | Scalar uncertainty score, $u(y) = \sqrt{\mathrm{tr}\, S(y)}$ |
| mError | The Euclidean (L2) norm between the predicted and ground-truth transformed positions, $\|\phi^{AB}(y) - \hat{\phi}^{AB}(y)\|_2$ with $y \in \Omega^B$, averaged over the ROI. |
| AURC | Area under risk–coverage curve, quantifies uncertainty–error trade-off |

Table 1: Summary of notations used throughout the paper.

# B APPENDIX: DETAILED PROOFS OF LEMMAS

**Preliminaries.** Recall the perturbed pair and composed output:

$$\hat{\phi}^{A'B}(y) = \left(\tau^{-1} \circ \phi^{AB}\right)(y) + \epsilon\tau(y), \qquad g_\tau(y) = \tau \circ \hat{\phi}^{A'B}(y) = \tau\left(\left(\tau^{-1} \circ \phi^{AB}\right)(y) + \epsilon\tau(y)\right),$$

with $\epsilon\tau(y) \sim \mathcal{N}\left(\mu_\epsilon(\tau; y), \Sigma_\epsilon(\tau; y)\right)$. Let $v_\tau(y) := (\tau^{-1} \circ \phi^{AB})(y)$ and denote the Jacobian of $\tau$ at $v(y)$ by $J_\tau(y) := D\tau|_{v(y)}$. *Definitions as in Sec. 3.2–3.3.* (See main text for Eq. (4), Eq. (7) and the error model.) [2]

PROOF OF LEMMA 3.1

For completeness, we restate Lemma 3.1 here:

> **Lemma 3.1.** Let the perturbed output be $g_\tau(y) = \tau\left((\tau^{-1} \circ \phi^{AB})(y) + \epsilon_\tau(y)\right)$, with $\epsilon_\tau(y) \sim \mathcal{N}(\mu_\epsilon(\tau; y), \Sigma_\epsilon(\tau; y))$. Without assuming any independence between $\tau$ and $\epsilon_\tau$ we have
>
> $$\mathbb{E}_\tau[g_\tau(y)] = \phi^{AB}(y) + \mathbb{E}_\tau[J_\tau(y)\mu_\epsilon(\tau; y)], \tag{16}$$
>
> $$\mathrm{Cov}_\tau[g_\tau(y)] = \mathbb{E}_\tau[J_\tau(y)\Sigma_\epsilon(\tau; y)J_\tau^\top(y)] + \mathrm{Cov}_\tau[J_\tau(y)\mu_\epsilon(\tau; y)]. \tag{17}$$

*Proof.* Let $v_\tau(y) := (\tau^{-1} \circ \phi^{AB})(y)$, so $\tau(v_\tau(y)) = \phi^{AB}(y)$. Apply a first-order Taylor expansion of $\tau$ at $v_\tau(y)$ to the argument $v_\tau(y) + \epsilon_\tau(y)$,

$$\tau(v_\tau(y) + \epsilon_\tau(y)) = \tau(v_\tau(y)) + J_\tau(y)\epsilon_\tau(y) + R_\tau(y),$$

where the remainder $R_\tau(y) = o(\|\epsilon_\tau(y)\|)$ is higher order in the perturbation. Because $\tau(v_\tau(y)) = \phi^{AB}(y)$, the offset simplifies and we obtain (to first order)

$$g_\tau(y) = \phi^{AB}(y) + J_\tau(y)\epsilon_\tau(y) + R_\tau(y).$$

Conditioning on $\tau$ and using the error model moments, $\mathbb{E}[\epsilon_\tau(y)\,|\,\tau] = \mu_\epsilon(\tau; y)$, $\mathrm{Cov}[\epsilon_\tau(y)\,|\,\tau] = \Sigma_\epsilon(\tau; y)$, we get

$$\mathbb{E}[g_\tau(y)\,|\,\tau] = \phi^{AB}(y) + J_\tau(y)\mu_\epsilon(\tau; y), \quad \mathrm{Cov}[g_\tau(y)\,|\,\tau] = J_\tau(y)\Sigma_\epsilon(\tau; y)J_\tau(y)^\top.$$

Apply the laws of total expectation and total covariance:

$$\mathbb{E}_\tau[g_\tau(y)] = \mathbb{E}_\tau\left[\mathbb{E}[g_\tau(y)|\tau]\right], \quad \mathrm{Cov}_\tau[g_\tau(y)] = \mathbb{E}_\tau\left[\mathrm{Cov}(g_\tau(y)|\tau)\right] + \mathrm{Cov}_\tau\left(\mathbb{E}[g_\tau(y)|\tau]\right),$$

which yield the stated formulas. Higher-order terms are neglected in the first-order approximation. □

PROOF OF LEMMA 3.2

For completeness, we restate Lemma 3.2 here:

> **Lemma 3.2.** Let $\tau(z) = Az + b$ be a random affine transformation with $A \in \mathbb{R}^{d \times d}$, $b \in \mathbb{R}^d$. Since the translation part cancels, $g_\tau(y) = \phi^{AB}(y) + A\epsilon_\tau(y)$. Then:
>
> $$\mathbb{E}_{A,b}[g_{A,b}(y)] = \phi^{AB}(y) + \mathbb{E}_A[A\mu_\epsilon(A; y)], \tag{18}$$
>
> $$\mathrm{Var}_{A,b}[g_{A,b}(y)] = \mathbb{E}_A[A\Sigma_\epsilon(A; y)A^\top] + \mathrm{Var}_A[A\mu_\epsilon(A; y)]. \tag{19}$$

*Proof.* Write $v_\tau(y) = (\tau^{-1} \circ \phi^{AB})(y)$. Since $\tau$ is affine,

$$g_\tau(y) = \tau(v_\tau(y) + \epsilon_\tau(y)) = A(v_\tau(y) + \epsilon_\tau(y)) + b = \underbrace{(Av_\tau(y) + b)}_{=\,\tau(v_\tau(y)) = \phi^{AB}(y)} + A\epsilon_\tau(y).$$

Thus $g_\tau(y) = \phi^{AB}(y) + A\epsilon_\tau(y)$ holds exactly (no linearization). Conditioning on $A$ (the distribution of $b$ cancels), we obtain $\mathbb{E}[g_\tau(y)\,|\,A] = \phi^{AB}(y) + A\mu_\epsilon(A; y)$ and $\mathrm{Cov}[g_\tau(y)\,|\,A] = A\Sigma_\epsilon(A; y)A^\top$. Unconditioning via total expectation/covariance gives the result. □

---

[2]We follow the composition and error setup in Section 3.2–Section 3: Eq. (4) and Eq. (7) define $\hat{\phi}^{A'B}$ and $g_\tau$, and the error model for $\epsilon\tau$.

PROOF OF COROLLARY 3.3

For completeness, we restate Corollary 3.3 here:

> **Corollary 3.3.** Let $\tau(z) = z + t$ with random $t$. Then $g_t(y) = \phi^{AB}(y) + \epsilon_t(y)$,
> and
>
> $$\mathbb{E}_t[g_t(y)] = \phi^{AB}(y) + \mathbb{E}_t[\mu_\epsilon(t; y)], \tag{20}$$
>
> $$\mathrm{Var}_t[g_t(y)] = \mathbb{E}_t[\Sigma_\epsilon(t; y)] + \mathrm{Var}_t[\mu_\epsilon(t; y)]. \tag{21}$$

*Proof.* For translations, $\tau^{-1}(z) = z - t$ and $\tau(v) = v + t$. Hence

$$g_t(y) = \tau\Big((\tau^{-1} \circ \phi^{AB})(y) + \epsilon_t(y)\Big) = \big(\phi^{AB}(y) - t + \epsilon_t(y)\big) + t = \phi^{AB}(y) + \epsilon_t(y),$$

exactly. Conditioning on $t$ yields $\mathbb{E}[g_t(y) \mid t] = \phi^{AB}(y) + \mu_\epsilon(t; y)$ and $\mathrm{Var}[g_t(y) \mid t] = \Sigma_\epsilon(t; y)$. Apply total expectation/variance to conclude. □

**Accuracy of the approximations.** The proofs for affine and translation perturbations are exact (no Taylor remainder). For general diffeomorphisms, Lemma 1 follows from a first-order expansion at $u(y)$; higher-order terms in $\epsilon_\tau(y)$ vanish as the perturbation scale decreases, and the result is accurate in the small-perturbation regime customarily used in practice.

**Why do we choose the additive error model instead of the compositional error model in Eq. (8)?** Registration error can be formulated compositionally, $\hat{\phi}^{AB} \circ (\phi^{AB})^{-1}$, or additively, $\hat{\phi}^{AB} - \phi^{AB}$. Both are mathematically valid: the compositional model reflects the geometry of diffeomorphic transformations, while an additive model reflects the standard regression view of predicting continuous-valued deformation fields. Our goal for the analysis is to rewrite the variance w.r.t. the error so that we can explain why the variance and error correlates and further interpret the uncertainty map. In practice, however, registration error is almost always computed additively (e.g., Euclidean distance between warped and target landmarks), which corresponds to $\hat{\phi}^{AB}(\cdot) - \phi^{AB}(\cdot)$. To keep the theoretical analysis consistent with the evaluation used in practice, we therefore adopt the additive error model.

# C APPENDIX: DETAILS OF THE EXPERIMENT SETTINGS

## C.1 IMPLEMENTATION OF THE TRANSFORMATION PERTURBATION

The transformations for the perturbation are sampled from the following distribution.

- Translation Perturbation: $t \sim \mathcal{U}(-t, t)$ with $t = 1\%$ of image shape;
- Shear Perturbation: shear factor sampled from $\mathcal{U}(-0.02, 0.02)$;
- Scale Perturbation: scale factor sampled from $\mathcal{U}(0.9, 1.1)$;
- B-spline transformation Perturbation (Deform): with grids distribution 10 pixels apart and displacement randomly sampled from $U(-12.5, 12.5px)$.

We sample $N = 50$ transformations for each perturbation type and compute the variance across all the experiments.

**Approach Statistics.** We report the memory usage and runtime overhead introduced by our uncertainty estimation to quantify its test-time cost. The measured runtime includes three components: loading the registration network, a one-time registration inference, and the computation of the uncertainty map with $N = 50$ perturbations (as described in Algorithm 1). For comparison, the baseline backbone inference time includes only model loading and a single registration pass, averaged over 50 runs to reduce variance. All experiments were conducted on an RTX 4500 Ada GPU. The results are summarized in Table 2. Note that perturbation transformations are generated on the CPU, which could be further optimized to reduce runtime.

| | uniGradICON | Uncertainty Estimation | | | |
|---|---|---|---|---|---|
| | | Transformation | Shear | Scale | Deform |
| Runtime (s) | 0.20 | 12.09 | 12.17 | 12.13 | 27.72 |
| GPU Memory (MB) | 2363 | 2630 | 2630 | 2630 | 2713 |

Table 2: Runtime and peak memory of the backbone model uniGradICON and our proposed uncertainty estimation approach.

## C.2 DATASETS, BACKBONES, AND EVALUATION METRICS

**Metrics.** We use three evaluation metrics: (1) Pearson correlation, (2) Spearman correlation, and (3) the area under the risk-coverage curve (AURC). The Pearson and Spearman correlations are computed between the uncertainty map and the voxel-wise registration error. These correlations are then averaged within each ROI and across the dataset to obtain instance-level and dataset-level correlation scores. The AURC is a scalar metric that quantifies the effectiveness of an uncertainty (or confidence) measure in identifying reliable predictions. It is computed from the risk–coverage curve, where coverage denotes the fraction of predictions retained after discarding the least confident ones, and risk denotes the corresponding prediction error on the retained set. By integrating risk over all coverage levels, the AURC summarizes how well uncertainty scores allow a trade-off between accuracy and coverage. For each voxel, the prediction error is computed as the Euclidean (L2) norm between the predicted and ground-truth transformed positions, $\|\phi^{AB}(y) - \hat{\phi}^{AB}(y)\|_2$ with $y \in \Omega^B$.

**Datasets.** In the experiments, we prepare three datasets that cover brain MRI, cardiac MRI, and abdomen CT. These datasets are used for quantitative evaluations. In addition, we conduct case studies on Brats-Reg (Baheti et al., 2021) and Learn2Reg ThoraxCBCT (Hugo et al., 2016; 2017) datasets.

*Curated Brain MRI.* We collect a brain MRI dataset by randomly sampling 20 MRI images from 11 brain MRI repositories (ABIDE (Di Martino et al., 2014), ADHD200 (Brown et al., 2012), ADNI (Jack Jr et al., 2008; Weiner et al., 2017), AIBL (Fowler et al., 2021), FreeSurfer (Fischl et al., 2002), COBRE (Mayer et al., 2013), Chinese-HCP (Vogt, 2023), HCP (Van Essen et al., 2012), ISBI2015 (Carass et al., 2017), MCIC (Gollub et al., 2013), OASIS3 (LaMontagne et al., 2019)), leading to a 220 MRI volumes dataset. This curated dataset consists of highly heterogeneous populations, including healthy subjects (HCP, Chinese-HCP), dementia (OASIS3, ADNI3, AIBL, FreeSurfer), Autism Spectrum Disorder (ABIDE), Attention Deficit Hyperactivity Disorder (ADHD), Schizophrenia (COBRE, MCIC), and Multiple Sclerosis (ISBI2015). We use the whole dataset in the evaluation with a real transformation experiment, and use 110 images for the simulated transformation experiment.

*ACDC Cardiac MRI.* We use the training set from the ACDC dataset (Bernard et al., 2018), which contains 100 pairs of cardiac MRI that are acquired at the end of the diastolic (ED) and systolic (ES) phase of the same subject.

*Learn2Reg Abdomen CT.* We use the training set of the Abdomen CT dataset (Xu et al., 2016) released by the Learn2Reg registration challenge, which contains 30 abdomen CT images. For the synthetic transformation experiment, we use the 30 images with synthetic transformations. In the real transformation experiment, we use the 30 images as the source image and randomly sample 30 images from the same list of images as the target image, composing the 30 pairs of images.

*IXI Brain MRI.* We use the IXI Brain MRI dataset following the preprocessing pipeline provided by TransMorph (Chen et al., 2022), and we adopt the same test split as in TransMorph. The processed dataset contains one atlas and 115 brain MRIs.

**Backbone registration models.** We use two publicly available registration networks as the backbone used to evaluate the confidence score measured by spatial perturbation.

*uniGradICON (Tian et al., 2024b).* It is a medical image registration foundation model that is trained across a curated heterogeneous dataset covering several anatomical structures. It demonstrates excellent generalization to both in-distribution and out-of-distribution registration tasks.

*SynthMorph (Hoffmann et al., 2021).* It is a modality-agnostic brain registration network that is trained purely on synthetic brain MRI and demonstrates great generalization and robustness for brain image registration. We also test this model on the cardiac and abdominal images. We note that SynthMorph provides both an affine model and a nonlinear model, and we use the nonlinear model in our experiments. We do not aim to measure the registration accuracy but the validity of the uncertainty estimation, namely, the correlation between the registration error and the estimated confidence score.

*TransMorph (Chen et al., 2022).* It is a transformer-based registration network that provides variant pre-trained models, including a probabilistic registration model trained with MC-dropout.

### C.3 Dataset-level Quantitative Analysis

In the main manuscript (Section 5.1), we provide a concise description of the experimental setup, present selected results, and summarize the main takeaways. In this appendix, we provide details of the experiment setting, report the complete results (including the ACDC cardiac MRI dataset), and offer an extended discussion.

**Experiment Setting.** To quantitatively evaluate the correlation between the registration error and the uncertainty estimated by our method, we simulate three types of deformation and apply the randomly simulated deformation to warp the image to get the paired image with ground truth deformation. We test the uncertainty estimation under the following three simulated ground truth deformations:

- Translation $t \sim \mathcal{U}(-t, t)$ with $t = 10\%$ of image shape;
- Affine $\tau(z) = Az + b$ with translation uniformaly sampled in $\mathcal{U}(-t, t)$ with $t = 10\%$ of image shape, shear factor sampled from $\mathcal{U}(-0.1, 0.1)$, and scale factor sampled from $\mathcal{U}(0.8, 1.2)$;
- Composition of two elastic B-spline transformations with grids distribution 10 pixels apart and displacement randomly sampled from $U(-12.5, 12.5px)^3$.

We use a composition of two elastic deformations in (iii) to mimic an extremely large deformation scenario (as shown in Fig. 4a).

While simulated transformations provide controlled conditions for evaluating registration uncertainty, they do not fully reflect the challenges in real registration scenarios. We design an experiment setting that approximates real-world conditions while providing access to ground-truth transformations. Specifically, we use a conventional registration algorithm (ANTs) (Avants et al., 2008) to estimate the transformations between real image pairs with affine followed by non-rigid registration. These estimated transformations are then applied to warp the source image, producing a new target image that shares the same transformation. Although this is a workaround, it ensures that the ground-truth transformations are derived from real registration problems, thereby closely mimicking practical registration scenarios while preserving access to reference transformations. We follow the same experimental design as in the simulated transformation study, applying different perturbation types (translation, affine, and nonrigid deformation) to evaluate both uniGradICON and SynthMorph.

**Extended discussion.** Fig. 7 reports additional results on the cardiac dataset. Overall correlation levels are lower than those observed for the brain and abdominal datasets. We hypothesize that this drop is related to the use of label maps of heart structures as the ROI mask: the small ROI size and limited texture variation within it produce uniformly high uncertainty values, which in turn weaken the correlation, as shown in Fig. 10.

In the following paragraph, we discuss the conditions under which strong correlations emerge and the scenarios where weaker correlations are expected.

**The intuition of why the perturbation variance relates to the error?** With the general form of the error model, for a single unperturbed prediction of the registration network, the *per-voxel*

---

[3] 12.5 pixel is the largest displacement one can for a B-spline transformation that has grids 10 pixels apart and a shape of $175 \times 175 \times 175$, which is the default shape we use in our experiments.

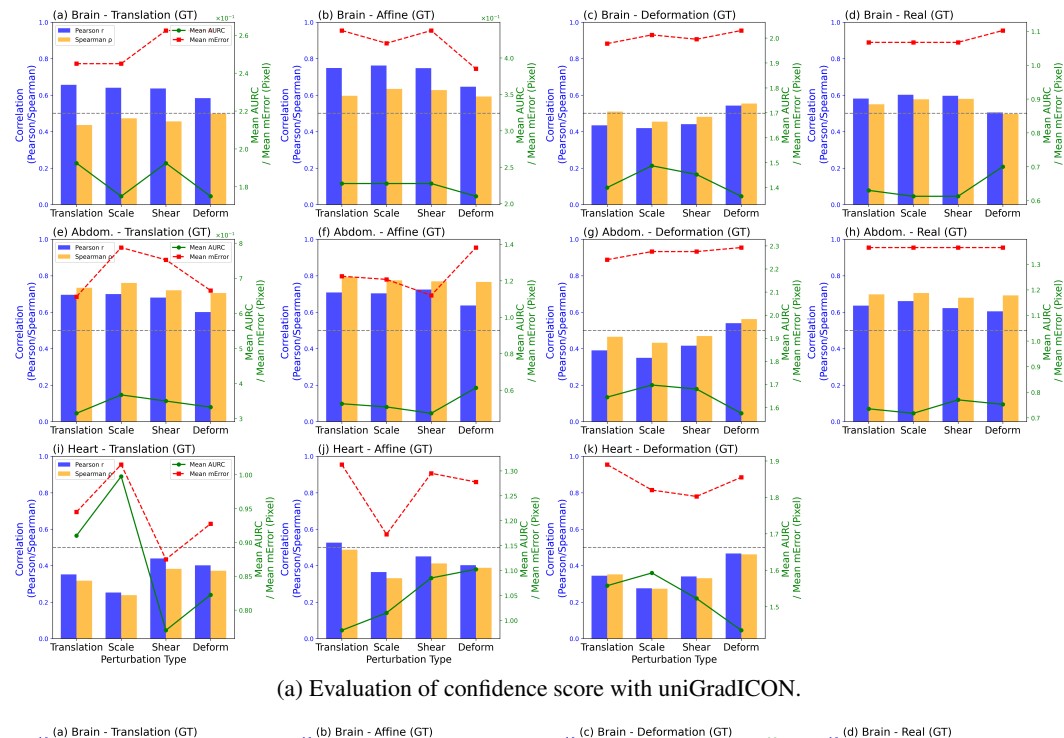

(a) Evaluation of confidence score with uniGradICON.

(b) Evaluation of confidence score with SynthMorph.

Figure 7: Correlation between the estimated confidence score and the true registration error for a combination of deformation types, dataset types, and perturbation types.

*registration error* decomposes into bias and variance

$$\underbrace{\mathrm{Error}(y)}_{\text{total error}} = \mathbb{E}\big[\|\epsilon(y)\|^2\big] = \|\mu_\epsilon(y)\|^2 + \underbrace{\mathrm{tr}\,\Sigma_\epsilon(y)}_{\text{shared term}}.$$

By contrast, the perturbation variance derived in Eq. (11) is

$$\underbrace{\mathrm{Var}_\tau\big[g_\tau(y)\big]}_{\text{uncertainty map}} = \underbrace{\mathbb{E}_\tau\big[J_\tau\,\Sigma_\epsilon(\tau;y)\,J_\tau^\top\big]}_{\text{shared term}} + \mathrm{Var}_\tau\big[J_\tau\,\mu_\epsilon(\tau;y)\big].$$

Both quantities contain the *mean intrinsic spread* $\mathbb{E}_\tau[J_\tau\Sigma_\epsilon J_\tau^\top]$. They differ in the other term:

$$\text{perturbation variance} \longleftarrow \mathrm{Var}_\tau\big[J_\tau\mu_\epsilon(\tau)\big] \quad \text{(how \textit{variance the centers are} moving)}$$

$$\text{total error} \longleftarrow \|\mu_\epsilon\|^2 \quad \text{(how \textit{far the centre is} from zero).}$$

When computing the correlation between the uncertainty map and the registration error, we will see the following results:

a) **High correlation** occurs when bias $\|\mu_\epsilon\|^2$ and its jitter $\mathrm{Var}_\tau\big[J_\tau\mu_\epsilon(\tau)\big]$ covary. Both maps rise where either component grows.

b) **Low or negative correlation** can arise in two scenarios: (i) when the dominant error is a perturbation-*invariant* bias, so the total error is dominated by a large $\|\mu_\epsilon\|^2$ term that the variance

cannot capture; (ii) when the network is highly uncertain about its prediction, producing a large $\mathrm{Var}_\tau\big[J_\tau\mu_\epsilon(\tau)\big]$, as observed in the cardiac dataset.

It is important to note that low correlation does not always imply that the uncertainty map is uninformative. Case (i) reflects a true underestimation, where the uncertainty fails to capture systematic bias. In contrast, case (ii) provides a valid signal of model indecision: even if the registration happens to be accurate, the high uncertainty correctly reflects low confidence and should still draw attention.

### C.4 Single-Case Quantitative Analysis

In the main manuscript (Section 5.2), we present a representative example to illustrate the behavior of the proposed uncertainty measure and highlight the main observations. In this appendix, we present quantitative results from a more randomly sampled subset of cases from the brain, abdominal, and cardiac datasets, and discuss the consistency of the findings.

### C.4.1 Additional Case for Brain

We include the result for another randomly sampled Brain MRI in Fig. 8 to demonstrate the generalization of our discussion.

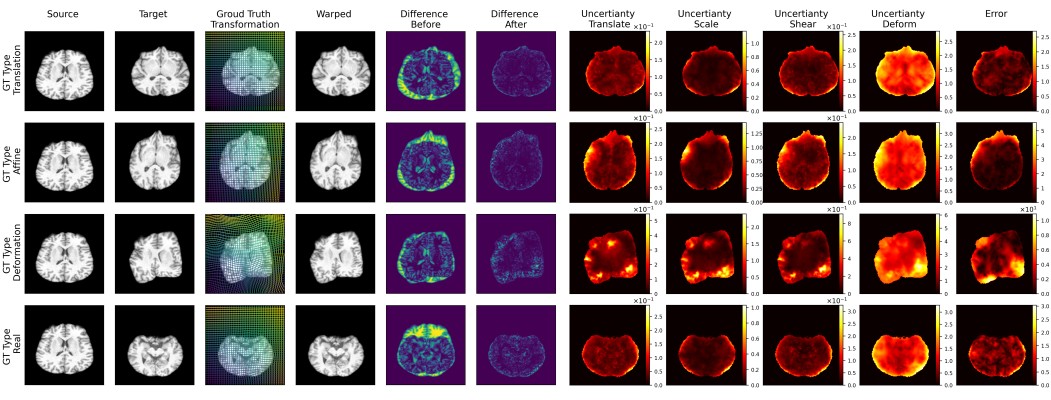

(a) The visualization.

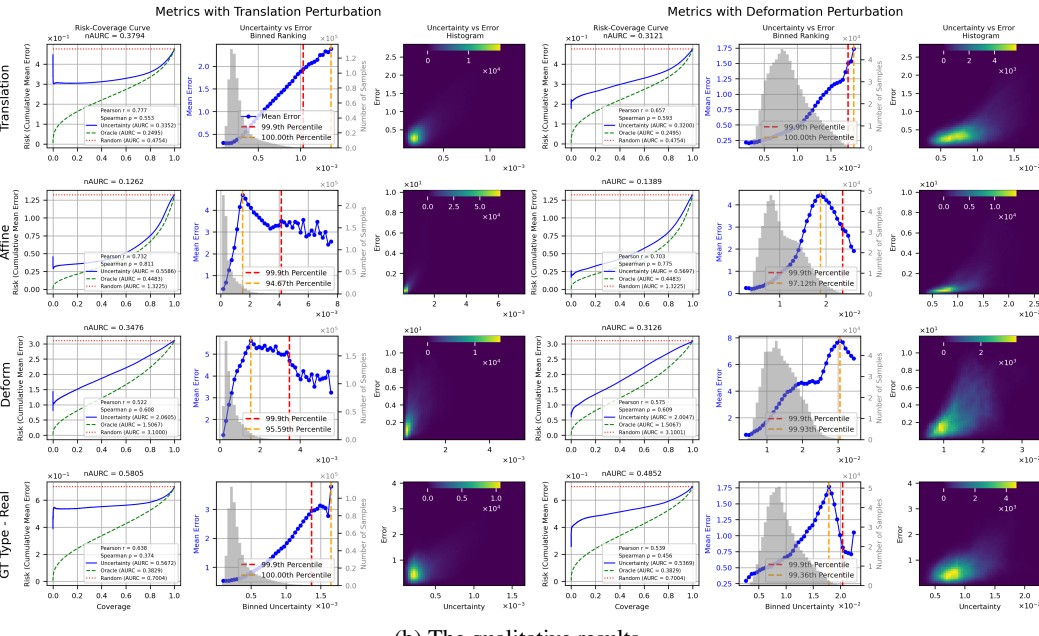

(b) The qualitative results.

Figure 8: The qualitative and quantitative results of the uncertainty map measured of a randomly sampled brain MRI image from **COBRE** for a combination of varying ground truth transformation types (GT), perturbation transformation types with uniGradICON.

### C.4.2 SINGLE CASE ANALYSIS FOR ABDOMEN

We conduct the same analysis as in Section 5.2 on one random abdomen CT image sampled from Learn2Reg Abdomen CT. The results are shown in Fig. 9a and Fig. 9b.

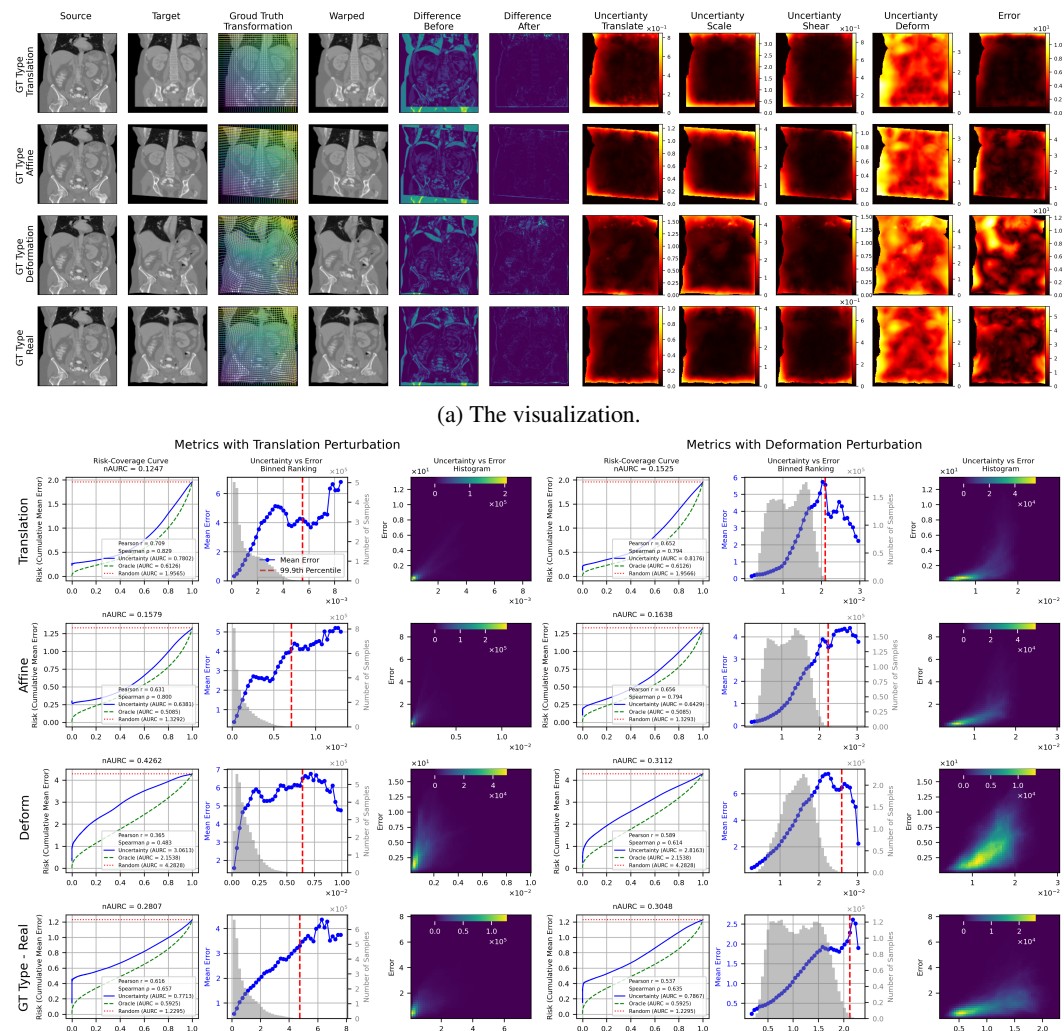

(a) The visualization.

(b) The quantitative results.

Figure 9: The qualitative and quantitative results of the uncertainty map measured of a randomly sampled abdomen CT image for a combination of varying ground truth transformation types (GT), perturbation transformation types with uniGradICON.

**Discussion.** Fig. 9 shows qualitative and quantitative results for one randomly sampled abdomen CT case. The qualitative visualization (Fig. 9a) demonstrates that the proposed uncertainty maps consistently highlight regions of misalignment. In addition, it also shows consistent observations as for the brain dataset:

- Error maps vary across ground-truth types (last column in Fig. 9a). The registration backbone tends to make localized registration errors when the ground truth transformation is nonlinear.
- Uncertainty maps vary across perturbation types under the same ground-truth transformation. As shown by the third row in Fig. 9a, linear perturbations often fail to reveal the localized errors (columns 7-9). Nonlinear perturbations distort the geometry more strongly and are therefore more effective at exposing unstable regions.

The quantitative evaluation (Fig. 9b) aligns with these qualitative findings. Translation-based perturbations achieve higher correlations with the true error (Pearson $r \approx 0.63$, Spearman $\rho \approx 0.47$) and lower nAURC values when the ground truth transformation is translation or affine, which is consistent with the qualitative evaluation in Fig. 9a. Moreover, for nonlinear ground truth transformations (third row in Fig. 9b), the binned error-vs-uncertainty curves of the translation perturbation flatten earlier than the deformation perturbation, indicating that many high-error voxels cannot be distinguished by uncertainty under linear perturbations, again showing that linear perturbations capture a different aspect of model instability.

### C.4.3 SINGLE CASE ANALYSIS FOR CARDIAC

We conduct the same analysis as in Section 5.2 on a randomly sampled cardiac MRI from the ACDC dataset. Results are presented in Fig. 10a and Fig. 10b. The ROIs, derived from segmentation maps of heart structures, are relatively small compared to the full image and contain limited texture variation. This lack of visual detail likely explains the elevated uncertainty observed in the maps, as the model faces inherent ambiguity in such low-texture regions. Consequently, high uncertainty arises even when registration error is limited, corresponding to scenario b)-(ii), where uncertainty reflects low model confidence rather than direct error magnitude.

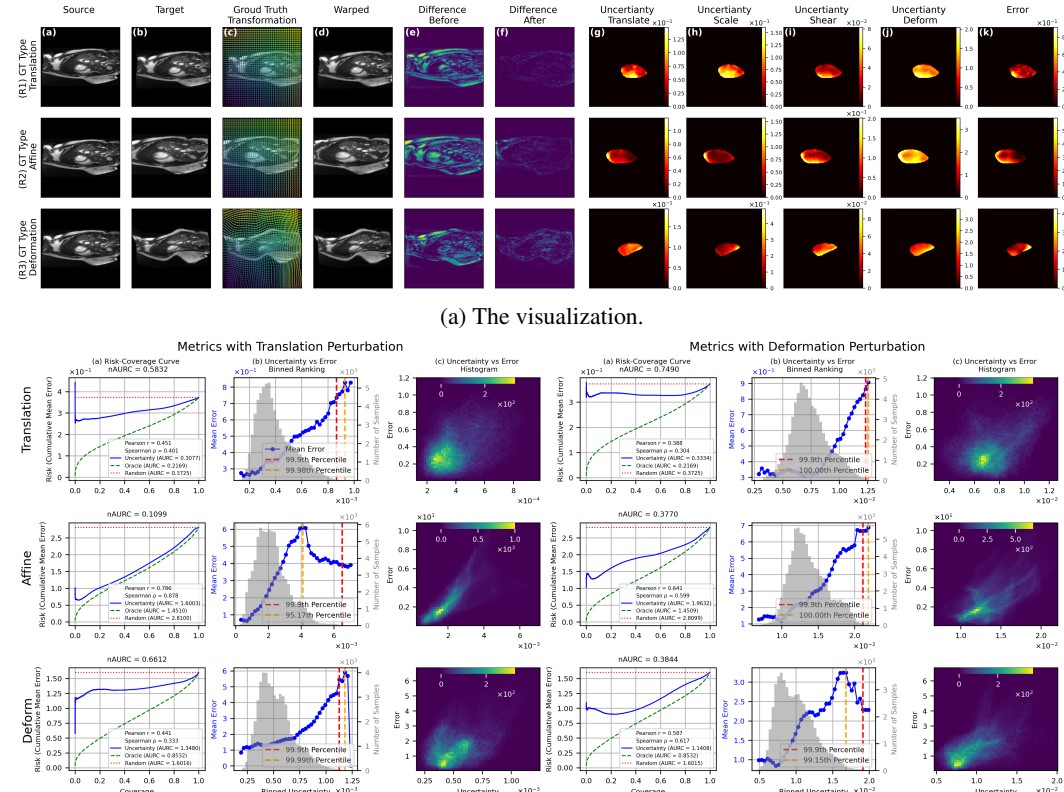

(a) The visualization.

(b) The quantitative results.

Figure 10: The qualitative and quantitative results of the uncertainty map measured of a randomly sampled cardiac MRI image for a combination of varying ground truth transformation types (GT), perturbation transformation types with uniGradICON.

## C.5 Case Study: Uncertainty Maps with Anatomical Inconsistencies

In the main manuscript (Section 5.4), we describe two representative case studies to demonstrate how the proposed uncertainty maps highlight clinically relevant inconsistencies, such as tumor re-section changes and field-of-view differences. In this appendix, we provide further details of the experimental design, including additional examples from the BraTS-Reg and Learn2Reg Thorax CBCT datasets, and offer a more extensive discussion of the observations. This experiment is run with uniGradICON.

### C.5.1 Case Study of the Pre-operative and Follow-up MRIs

The first case study investigates whether the uncertainty map can identify inconsistent anatomical structures, such as pathological changes in the brain, before and after tumor resection. Beyond the example shown in the main manuscript (Fig. 6a), we additionally sample five pairs of pre-operative and follow-up MRIs from the BraTS-Reg dataset (Baheti et al., 2021) and visualize the results in Fig. 11. Registrations and corresponding uncertainty maps are computed from T1-weighted images, while T2-weighted images are also included to better delineate abnormal regions. The results show that the uncertainty maps consistently highlight tumor-related inconsistencies in Cases 023, 030, 034, and 080, in line with the observations reported in Section 5.4. In contrast, Case 037 exhibits no prominent high-uncertainty regions, which can be attributed to the strong visual similarity between its source and target images. Taken together, these results confirm that the proposed uncertainty map generalizes beyond a single example and can reliably flag pathological inconsistencies across different subjects.

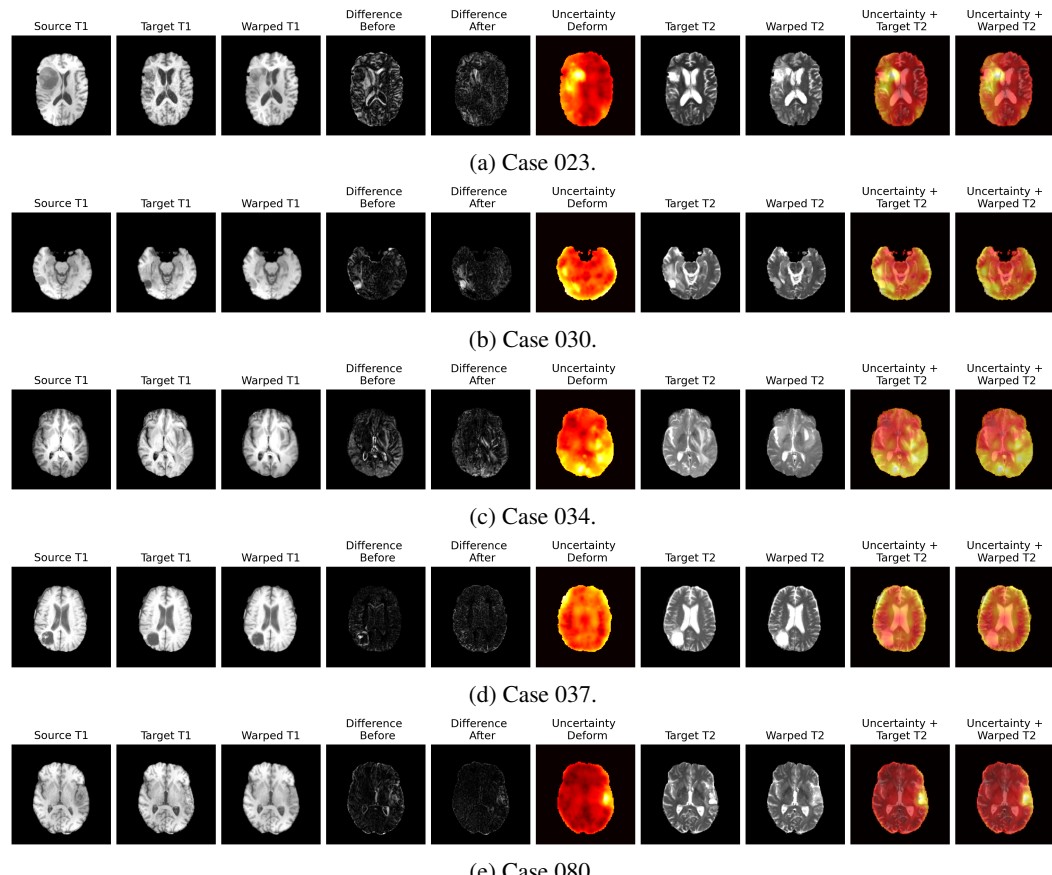

(a) Case 023.

(b) Case 030.

(c) Case 034.

(d) Case 037.

(e) Case 080.

Figure 11: Uncertainty map estimated during the registration between pre-operative and follow-up MRIs.

### C.5.2 CASE STUDY OF CBCT AND CT WITH DIFFERENT FOVS

We present the results for three additional pairs of images from the Learn2Reg ThoraxCBCT dataset (Hugo et al., 2016; 2017) with the same experiment setting in Section 5.4. As shown in Fig. 12, the CBCT/CT registrations consistently exhibit elevated uncertainty in the truncated regions above and below the lung, whereas the control pairs ($CT'$/CT) exhibit uniformly low uncertainty. This pattern is observed across all three tested pairs, reinforcing the conclusion drawn in Section 5.4: the proposed uncertainty map reliably highlights regions of anatomical inconsistency caused by differences in FOV, and the finding generalizes beyond the specific examples shown in the main manuscript.

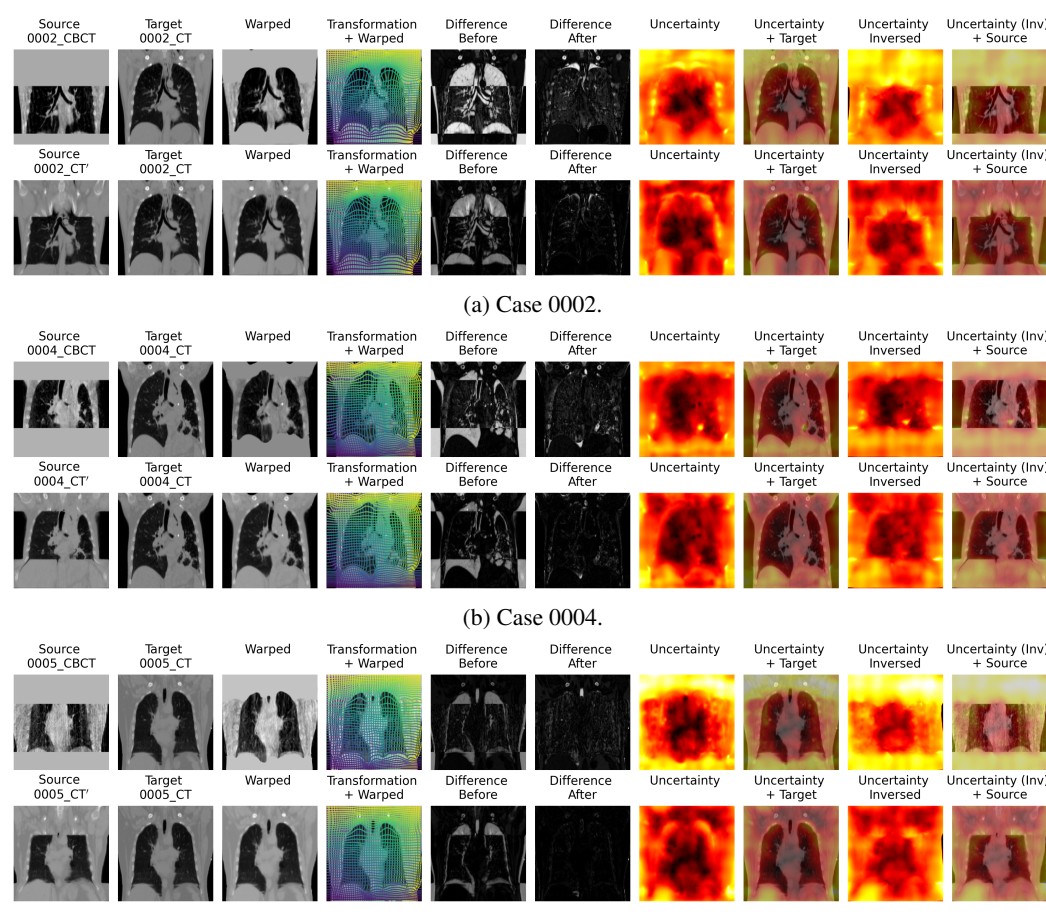

(a) Case 0002.

(b) Case 0004.

(c) Case 0005.

Figure 12: Uncertainty map estimated during the registration between CBCT and CT, which have different fields of view.

