# OpenReview forum: "Test-time Uncertainty Estimation for Medical Image Registration via Transformation Equivariant"
_ICLR.cc/2026/Conference — Submitted to ICLR 2026_

### Official Review · Reviewer_Edea · 2025-10-25

**Soundness:** 3
**Presentation:** 3
**Contribution:** 3
**Rating:** 8
**Confidence:** 4

**Summary:**

This paper presents a test-time, model-agnostic framework for estimating uncertainty in deep medical image registration networks. The core idea exploits the transformation equivariance property which means that for a true registration mapping, spatial perturbations of the input should produce consistent transformations. The authors show that deviations from this equivariance can be used to estimate uncertainty without retraining or modifying the model. The authors further provide a theoretical decomposition of the approach and evaluate it across multiple datasets and registration models. Experimental results demonstrate a strong correlation between the proposed uncertainty maps and the true registration errors.

**Strengths:**

1. Good novelty: The paper provides the first clear theoretical derivation of perturbation-based uncertainty estimation specifically tailored for image registration. The proposed formulation naturally aligns with the equivariance property inherent to the registration task.

2. Strong theoretical grounding: The authors decompose perturbation-induced variance into intrinsic spread and bias jitter. This formalism connects geometric consistency with probabilistic uncertainty and bridge the conceptual gap in registration theory.

3. Solid experimental validation: The authors perform comprehensive experiments across multiple datasets (brain, cardiac, abdominal, lung) and different pretrained models (uniGradICON, SynthMorph) under various perturbations (translation, affine, deformation). Across all settings, the uncertainty maps show strong correlation with true registration errors, and the method successfully identifies field-of-view (FOV) mismatches and anatomical inconsistencies, demonstrating both robustness and clinical relevance.

**Weaknesses:**

1. Lack of comparison with existing uncertainty estimation methods: The evaluation primarily focuses on the proposed framework without direct quantitative comparison to established uncertainty estimation approaches (e.g., Bayesian inference, MC dropout, deep ensembles, or variational registration models). While the proposed method demonstrates strong correlation between uncertainty maps and true registration errors, it remains unclear whether it outperforms these existing techniques. Including such baselines would strengthen the empirical evidence for the claimed advantages in both accuracy and practicality.

**Questions:**

1. Could you please provide a comparison with other uncertainty estimation methods?

2. Could you add color bars in Figure 5, similar to those in Figure 4?

---

> ### Author Response · Authors · 2025-11-20
>
> We thank the reviewer for the thoughtful and encouraging evaluation of our work, especially the recognition of (i) the theoretical grounding of our transformation-equivariance formulation, (ii) the decomposition into intrinsic spread and bias jitter, and (iii) the comprehensive experimental validation across datasets, anatomical regions, and pretrained models.
>
> ---
>
> ***Q1. Comparison with existing uncertainty estimation methods***
>
> Following the reviewer’s suggestion, we have added a new experiment in Sec. 5.3, comparing our method against uncertainty estimated via MC dropout. Specifically, we use the publicly available TransMorph-Bayes model as the backbone and evaluate uncertainty under three configurations:
> 1. **MC-dropout UE**: TransMorph-Bayes with dropout **enabled**
> 2. **Ours (static)**: Our method with dropout **disabled** TransMorph-Bayes
> 3. **Ours (total)**: Our method with dropout **enabled** TransMorph-Bayes
>
> This design isolates epistemic uncertainty (via dropout) from perturbation-based uncertainty (our test-time method). Across all three settings, our uncertainty maps show strong correlation with the MC-dropout baseline, indicating that our perturbation-induced variance captures meaningful predictive variability.
> | Perturbation type |  MC-dropout vs Our (static) | MC-dropout vs Our (total) |Our (static) vs Our (total)|
> |---------------------|:--------:|:--------:|:--------:|
> | Translation         |   0.669  |   0.672  | 0.977 |
> | Deformation      |   0.891  |   0.940  | 0.943 |
>
> Importantly, our framework is model-agnostic: it applies directly to both deterministic and probabilistic registration networks without retraining, as discussed in Sec. 3.3 (“How does the error model affect the variance?”). Our method serves as a complement or enhancement of existing uncertainty estimation approaches, rather than a replacement for Bayesian approaches.
>
>
> ***Q2. Color bars in Figure 5***
>
> We have updated Figure 5 (Figure 6 in the revised manuscript) to include color bars consistent with Figure 4.
>
> ---
>
> We thank the reviewer again for their constructive feedback. The added comparison experiment and improved visualization directly address the reviewer’s questions and further strengthen the contribution and clarity of the work.

---

> ### Author Response · Authors · 2025-11-24
> **Follow-up on the Discussion**
>
> Dear Reviewer Edea,
>
> Thank you again for your thorough review and the positive feedback. We hope our responses have fully addressed your concerns.
>
> Please kindly let us know if you have any remaining questions or concerns.
>
> Best,
>
> Authors of paper #6128

---

### Official Review · Reviewer_1SzL · 2025-10-26

**Soundness:** 3
**Presentation:** 3
**Contribution:** 3
**Rating:** 6
**Confidence:** 3

**Summary:**

This paper presents a test-time uncertainty estimation framework for deep medical image registration based on transformation equivariance. The method leverages spatial perturbations applied to input images and measures the resulting variability in the predicted deformation fields to quantify uncertainty. It introduces a theoretical decomposition of this variability into intrinsic spread and bias jitter, corresponding to different sources of uncertainty. The approach is model-agnostic and can be applied to any pretrained registration network without retraining. Experiments on multiple anatomical datasets and architectures show that the proposed uncertainty maps correlate well with registration errors and highlight anatomically inconsistent regions.

**Strengths:**

- The framework provides a clear and general way to estimate uncertainty at test time, requiring no modification or retraining of existing registration models.
- The authors evaluate the method across several datasets, anatomical regions, and architectures, showing strong empirical correlation between predicted uncertainty and registration errors.

**Weaknesses:**

- **Empirical nature of the uncertainty measure**: The proposed uncertainty reflects the variability of the warped output under spatial perturbations (i.e., uncertainty over $g(y)$), rather than directly capturing the model’s epistemic uncertainty. While the authors demonstrate a strong empirical correlation between this variance and registration error, the theoretical connection remains somewhat informal and could be further clarified.

- **Simplifying additive error assumption**: Equation (1) assumes that the model’s prediction error is additive with respect to the deformation field. This assumption may not hold in nonlinear or diffeomorphic registration, where transformation errors are typically compositional rather than additive, raising concerns about the theoretical validity of the variance decomposition.

- **Limited technical novelty**: The idea of using test-time perturbations for uncertainty estimation is not new and has been applied in other domains such as classification and segmentation. This work primarily adapts the concept to registration rather than introducing a fundamentally novel methodology.

**Questions:**

Please see the weakness section.

---

> ### Author Response · Authors · 2025-11-20
>
> We thank the reviewer for acknowledging our effort to develop a general, ready-to-deploy uncertainty estimation method for existing registration models.
>
> -------
>
> ***Q1. Empirical nature of the uncertainty measure***
>
> We thank the reviewer for the thoughtful feedback and for highlighting the strengths of our empirical results. We agree that our test-time variance is not a direct Bayesian estimate of epistemic uncertainty. However, as the reviewer noted, it correlates strongly with registration error in practice. Our decomposition is designed to explain why this empirical correlation holds.
> The total registration error decomposes as
> $$\\underbrace{\\varepsilon(y)}\_{\\text{total error}}=\\mathbb{E}[\|\|\\epsilon(y)\|\|^{2}]=  \|\|\\mu_{\\epsilon}(y)\|\|^{2}+\\underbrace{\\operatorname{tr}\\Sigma_{\\epsilon}(y)}\_{\\text{shared term}},$$
>
> and our proposed variance decomposes as
>
> $$\\underbrace{\\mathrm{Var}\_{\\tau}[g\_{\\tau}(y)]}\_{\\text{uncertainty map}}=\\underset{\\text{shared term}}{\\underbrace{\mathbb{E}\_{\\tau}[J\_{\\tau}\\Sigma\_{\\epsilon}(\\tau;y)J\_{\\tau}^{\\top}]}}+\\mathrm{Var}\_{\\tau}[J\_{\\tau}\\mu\_{\\epsilon}(\\tau;y)].$$
>
> Both quantities contain the same variability term. The remaining terms $\|\\mu_{\\epsilon}(y)\|^{2}$ (systematic bias) and $\\mathrm{Var}\_{\\tau}[J\_{\\tau}\\mu\_{\\epsilon}(\\tau;y)]$ (sensitivity of the bias to perturbations) are the reasons why these two measured quantities differ.
>
> The correlation between variance and error is strong when these two terms co-vary **(1)**, consistent with the general empirical observation (less stable predictions tend to have larger systematic errors). Conversely, the correlation may weaken when
>
> **(2.1)** large $\\mu\_\\epsilon$ but small $\\mathrm{Var}\_{\\tau}[J\_{\\tau}\\mu\_{\\epsilon}(\\tau;y)]$: the model consistently makes the same mistakes, or
>
> **(2.2)** small $\\mu\_\\epsilon$ but large $\\mathrm{Var}\_{\\tau}[J\_{\\tau}\\mu\_{\\epsilon}(\\tau;y)]$: near correct but unstable predictions.
>
> Importantly, in scenarios of (1) and (2.2), a large variance either signals large error or low reliability, which is precisely the intended use of our method for downstream analysis. We clarified the above discussion in Appendix C.3.
>
> ***Q2. Simplifying additive error assumption***
>
> We thank the reviewer for raising this thoughtful theoretical point. Registration error can be formulated compositionally,
> $\hat{\phi}^{AB} \circ (\phi^{AB})^{-1},$
> or additively,
> $\hat{\phi}^{AB} - \phi^{AB}.$ Both are mathematically valid: the compositional model reflects the geometry of diffeomorphic transformations, while an additive model reflects the standard regression view of predicting continuous-valued deformation fields. Our goal is to undertand how the variance relates to the error, i.e., to rewrite
> $$\\mathrm{Var}= g(\\hat{\\phi}^{AB} \\circ (\\phi^{AB})^{-1})$$
> or
> $$\mathrm{Var}= g(\hat{\phi}^{AB} - \phi^{AB})$$
> so that we can explain why variance and error correlate and how to interpret variance in practice.
>
> In practice, however, registration error is almost always computed additively (e.g., Euclidean distance between warped and target landmarks, which corresponds to $\hat{\phi}^{AB}(\cdot) - \phi^{AB}(\cdot)$. **To keep the theoretical analysis consistent with empirical evaluation, we therefore adopt the additive error model**.
>
>
> Furthermore, our uncertainty estimator is designed to be model-agnostic and does not assume that predicted transformations are diffeomorphic. In realistic settings, networks may not enforce diffeomorphism, and even diffeomorphic architectures can yield non-diffeomorphic outputs under poor training (e.g., early stop during training or inappropriate hyperparameters). A compositional error model would apply only to a restricted set of ideal settings, whereas the additive model holds universally.
> For these reasons, we use the additive error model: it is the most general, empirically consistent, and practically interpretable formulation for our analysis.
>
> ***Q3: Limited technical novelty***
>
> We thank the reviewer for raising this point. Although our method involves applying spatial perturbations at test time, it is not derived from test-time augmentation techniques. Instead, the formulation comes directly from the transformation-equivariance property of registration, which leads to a task-specific way of probing the model’s geometric consistency. To our knowledge, no prior work has developed an equivariance-derived uncertainty framework for registration or provided a theoretical decomposition linking perturbation-induced variance to registration prediction error. This decomposition offers new insight into when uncertainty reflects error and how to interpret uncertainty maps in practice. Finally, our approach is practically valuable: it enables uncertainty estimation for any pretrained registration model across architectures and anatomies without retraining or architectural modifications.

---

> ### Author Response · Authors · 2025-11-24
> **Follow-up on the Discussion**
>
> Dear Reviewer 1SzL,
>
> Thank you again for your thorough review and the positive feedback. We hope our responses have fully addressed your concerns.
>
> Please kindly let us know if you have any remaining questions or concerns.
>
> Best,
>
> Authors of paper #6128

---

### Official Review · Reviewer_3ar4 · 2025-11-01

**Soundness:** 3
**Presentation:** 3
**Contribution:** 2
**Rating:** 6
**Confidence:** 2

**Summary:**

The paper proposed to estimate image registration uncertainty, given equivariance under deformations.
In my understanding, registration solutions are perturbed, then uncertainty is estimated in terms of intrinsic spread and bias jitter terms, which correspond to mean and variance. Experiments demonstrate the method on images of brain, heart, thorax, abdominal regions, MRI and CT modalities.

**Strengths:**

The method can be applied to the output of registration algorithm.

Uncertainty maps remain relatively consistent across linear perturbations, less so for deformable perturbations.

Uncertainty appears highest in image regions containing occlusion or structure present in only one image, e.g. brain tumors or variable field of view, this is encouraging and aligns with intuition.

**Weaknesses:**

Regarding identifying high uncertainty in occlusions, pathology, however it’s not necessarily difficult for a naive image differencing or intensity variance estimation to pick up such patterns.

Regarding graphs in Figure 6, 8 b), I do not have a strong intuition as to how meaningful or useful these are.

The literature seems to be lacking, the Gaussian Process model is has been used to estimation medical image registration uncertainty in a very similar way.
Wassermann, Demian, et al. "Probabilistic diffeomorphic registration: representing uncertainty." International Workshop on Biomedical Image Registration. Cham: Springer International Publishing, 2014.

**Questions:**

See previous section

---

> ### Author Response · Authors · 2025-11-20
>
> We thank the reviewer for the constructive feedback and for acknowledging the strengths of our framework, including its applicability to existing registration models and its ability to highlight anatomically inconsistent regions. We appreciate the reviewer’s interest in understanding the intuition behind our empirical analyses and address each point below.
>
> ------
>
> ***Q1: On Whether Image Differencing or Intensity-Based Criteria Can Detect Occlusions or Pathology***
>
> We thank the reviewer for raising this point. While naive image differencing can detect obvious intensity mismatches, it does not reliably identify registration failures, and importantly, it can only be applied after registering two images. As shown in Fig. 5(b), even when part of an anatomy is missing in one image (e.g., the lower lung region is absent in the source), the registration network often still aligns the visible boundary. The warped images may therefore appear well-aligned, making it difficult, often impossible, to judge correctness from image differencing or intensity variance alone without manual inspection.
>
> Furthermore, in large-scale clinical datasets, low-quality and outlier cases are highly diverse: motion artifacts, partial FOV, missing structures, unusual pathology, inconsistent positioning, and scanner artifacts. It is not feasible to enumerate all such cases or design manual intensity-based criteria that reliably detect each scenario after registration.
>
> In contrast, our uncertainty map summarizes all these diverse failure modes into a single quantity, the stability of the predicted deformation under task-preserving perturbations. This allows us to flag unreliable regions even when the images themselves appear aligned. Such a capability is particularly important for large-scale, data-centric pipelines, where manual QC of thousands of 3D registrations is infeasible. Our method therefore provides a practical and general QC mechanism for pretrained registration models already deployed in existing pipelines.
>
> ***Q2: Intuition Behind Figures 6 and 8(b) and Their Practical Meaning***
>
> We thank the reviewer for bringing up this question. Figures 7 and 9 (Figure 6 and 8 in the original manuscript) analyze how the proposed variance reflects registration error under different deformation regimes. Figure 7 examines this relationship at the dataset level. We observe strong correlation overall, but with notable variation between linear and nonlinear ground-truth transformations, suggesting that different perturbations probe different aspects of the model’s geometric stability.
>
> To understand this variation, Figure 9(a) visualizes error and uncertainty maps under different settings:
> 1. Error maps vary across ground-truth types. The registration backbone tends to make localized registration errors when the ground truth transformation is nonlinear (last column in Figure 9(a)).
> 2. Uncertainty maps vary across perturbation types under the same ground-truth transformation. As shown by the third row in Figure 9(a), linear perturbations often fail to reveal the localized errors (columns 7-9). Nonlinear perturbations distort the geometry more strongly and are therefore more effective at exposing unstable regions.
>
> Figure 9(b) provides quantitative confirmation at the instance level. For nonlinear ground-truth transformations (third row in Figure 9(b)), the binned error-versus-uncertainty curves flatten earlier under linear perturbations than under nonlinear perturbations, indicating that many high-error voxels cannot be distinguished by uncertainty under linear perturbations, again showing that linear perturbations capture a different aspect of model instability.
>
> Overall, Figures 7 and 9 demonstrate that uncertainty and error are strongly correlated in most settings, and that linear and nonlinear perturbations reveal complementary types of instability, with nonlinear perturbations better detecting localized failures. We revised the manuscript to clarify this intuition.
>
> ***Q3: Relation to Prior Probabilistic Registration Methods***
>
> We thank the reviewer for bringing this work to our attention. We now discuss it alongside the other related probabilistic approaches already included in our manuscript (e.g., line 48, Risholm et al., Simpson et al.), all of which introduce stochastic components into the transformation model and estimate or sample from a posterior over deformation parameters. These methods fundamentally rely on probabilistic modeling of the deformation space.
>
> ----
>
> We thank the reviewer again for the thoughtful comments. We incorporate clarifications regarding the intuition behind Figures 7 and 9 and refine our explanations to improve accessibility. Overall, we believe the empirical and theoretical analyses together demonstrate that the proposed equivariance-based variance provides meaningful and practical reliability information for a wide range of pretrained registration models.

---

> ### Author Response · Authors · 2025-11-24
> **Follow-up on the Discussion**
>
> Dear Reviewer 3ar4,
>
> Thank you again for your positive feedback. We hope our responses have fully addressed your concerns.
>
> Please kindly let us know if you have any remaining questions or concerns.
>
> Best,
>
> Authors of paper #6128

---

### Official Review · Reviewer_4VP3 · 2025-11-05

**Soundness:** 1
**Presentation:** 2
**Contribution:** 1
**Rating:** 0
**Confidence:** 4

**Summary:**

The paper deals with the uncertainty of image registration methods that are typically ill posed and have a multitude of possible solutions. Instead of standard Bayesian networks that require architectural changes, the method aims to use equivariances of registration to use a test-time augmentation of the inputs to estimate the variance of the produced deformation fields. The paper estimates the variance of the model under different perturbations of the input which is decomposed into two terms: an intrinsic "spread", and a "bias jitter".

**Strengths:**

1. Evaluation is performed with two state-of-the-art methods on multiple organs, spatial dimensions, and imaging modalities. The experiment  setup is well motivated for the task that is being tackled , i.e. uncertainty estimation. Applications like registration with limited FOV and tumor registration are good motivating examples of showing the impact of a good uncertainty registration method.

2. The method proposes estimation of a "perturbation-based uncertainty" given a model and . This is a research area that is critical in calibrating predictions by the amount of uncertainty due to the illposedness of the data. Registration is typically highly overparameterized, even with regularization, with potentially infinite compatible displacement fields that satisfy the image similarity constraint. This implies a high amount of aleatoric uncertainty due to the nature of the problem, which can form an excellent basis for imbuing models with the capacity to estimate this uncertainty. This uncertainty can also allow a user to perform guided registration in areas where the aleatoric uncertainty is high (i.e. large number of solutions are possible).

**Weaknesses:**

1. The interpretation of performing test-time augmentation to perform some kind of  uncertainty estimation deviates from the interpretation in green box in Line 192. The latter interpretation implies that a "true deformation" is equivariant to linear or nonlinear diffeomorphic perturbations but the network may not be - essentially measuring the equivariance error of the model. For a (hypothetical) model that does respect the equivariance of the task and somehow produces transforms consistent with the applied transform $\tau$, does it imply that the model has no estimates of uncertainty? I would argue that it is not true - as the ill-posedness of the problem does not disappear if the model becomes aware of the equivariance of the problem (and by extension the aleatoric uncertainty remains unchanged) . So the formulation is **not** measuring aleatoric uncertainty since the "equivariance error" can be reduced in the model by training it to respect the constraint in Equation (7) but the aleatoric uncertainty is intrinsic to the datum and cannot be reduced. On the other hand, epistemic uncertainty estimates the variance of the output (more below) under a posterior distribution over the model parameters. The proposed idea works with a pretrained network and does not impose any distribution over the model weights - so the proposed method does not compute epistemic uncertainty either. To me, this is a major fundamental issue with the paper - a simple argument showing that the proposed method captures neither epistemic or aleatoric uncertainty - and I would urge the authors to discuss and clarify this in the rebuttal.
2. Math in the paper is not rigorous at all (see concerns below). There is no such thing as "perturbation-based uncertainty". If new concepts are introduced, they must be compared and contrasted with more established concepts like uncertainty like aleatoric, epistemic, and predictive uncertainties. A highly theoretical area of work like uncertainty estimation requires very rigorous treatment of the proposed method.
3. A lot of terms are used very colloquially - e.g. in Line 39 "they predict a deformation field from an image pair, but not an estimate of how trustworthy that field is" - what is the scope of a "trustworthy field"? Is it a prediction with low uncertainty estimates?
4. Line 47 - "While effective, these approaches are computationally expensive and do not scale well." - any method that aims to estimate epistemic uncertainty will be computationally expensive for two reasons - (1) the epistemic uncertainty is a measure of the variance of the posterior $Var_{P(W|D)}[E_{P(Y|X;W)}Y]$ where $W$ are the weights of the network, and $D$ is the dataset, $X, Y$ are the inputs and outputs. Since the expectation or variance under the posterior typically does not have a closed form solution, one has to compute the empirical MC estimates [1, 2]. There is no other way to estimate model uncertainty unless a closed form solution can be derived. The other way to compute epistemic uncertainty cheaply is to choose an "restricted prior" $P(W)$ (delta distribution on the backbone layers from a pretrained network and a regular prior on the final layer) - but that is not interesting enough (similar to linear probing if the backbone feature extractor is robust enough)
5. Line 88: "We present the first theoretical account of perturbation-based uncertainty in image registration, showing how perturbation-induced variance decomposes into intrinsic spread and bias jitter" - this is an incorrect statement. Test-time augmentation has been around in the literature for a long time - the paper happens to perform test-time augmentation with invertible linear (rigid, affine) or nonlinear (spline) deformations. This claim needs to be rewritten to highlight the connections of the perturbation-based uncertainty to test-time augmentation and why this particular test-time augmentation makes sense for the considered task.
6. Line 116: "However, few works have explored uncertainty estimation in medical image registration." - no citations! Discussion with these other methods and how they are different from the proposed work are key pieces of related work missing from the paper.
7. Line 131 "Test time augmentation fundamentally measures something different compared to the bayesian deep learning frameworks. It is important to distinguish these  concepts instead of clubbing them together" - this is indeed the case because test-time approaches measure something that is indicative of aleatoric uncertainty rather than epistemic. If the input contains intrinsic uncertainty (e.g. registration task can have multiple solutions) regardless of the model, it falls under the purview of aleatoric uncertainty. These concepts must be discussed more holistically in the paper.
8. shouldn't the "error model" be compositive instead of additive - $\epsilon(y)$ captures the possible space of diffeomorphic perturbations that do not decrease the image similarity cost function (e.g. local diffeomorphic flows within subcortical regions). More formally, consider the support set of $\epsilon(y)$, i.e. $S_e(I_f, I_m) = \{\epsilon(y): Sim(I^A\circ (\phi \circ \epsilon), I^B) \ge Sim(I^A\circ (\phi), I^B)\}$ that tells us all the admissible perturbations that can be applied without decreasing the image similarity - these lead to amiguity of the solution. For example, if $S_e = \phi$ (empty set), then only a single minimizer exists, and there should be no uncertainty about which deformation maximizes image similarity. Using a well-defined measure of the support set, i.e. $\int_{S_e} d\mu(\epsilon)$ is an estimate of the amount of perturbations that can be applied. This is not a variance by itself, but is a meaningful proxy that quantifies the "amount of ill-posedness". An additive loss will allow inclusion of non-diffeomorphic updates that satisfy the similarity constraint, but a compositive "error term" will possibly make analysis difficult.
9. Lines 238-244 : There is no epistemic uncertainty in the blue arrow term. Any amount of mathematical connection between this term and how it relates to a standard definition of epistemic uncertainty. A similar statement for the purple arrow. This ties back to my feedback on the rigor of math (Point 2) and the argument that the quantity captures neither epistemic nor aleatoric invariance (Point 1).
10. "Bias jitter" is a slightly misleading term (although I understand where the naming comes from - a "jitter/variance" of the bias term) but it can be potentially misleading - The second term in Equation (9) is not a bias either, it is also a covariance term. I'd recommend choosing a name that makes more sense. Jitter is a colloquial term and can mean different things depending on context [3].
11. Figure 3: In contrary to what the paper says, there are actually no meaningful trends between the PCC / SC and the performance. For each dataset, the mean error is also not high or low for any dataset / perturbation combination.


Minor nits:
1. Line 36: "Modern learning based approaches can now deliver real-time registration that rivals or surpasses classical methods." - add citations to support this statement. Also see CLAIRE, FireANTs, NiftyReg and ConvexAdam for counter-examples to this statement.
2. Line 50: "but they typically underestimate uncertainty, fail to capture the spatial structure of deformation fields, and remain tightly coupled to specific architectures" - this is not a limitation of the variational formulation but the chosen priors on the model. The phrase about "failure to capture spatial structure of deformation fields" needs to be removed or supported with related literature.
3. Literature on classical image registration and their strengths and weaknesses is totally missing from the paper. Since the model only applies test-time augmentations regardless of models, it should be applicable to iterative methods as well. Do these methods show different behavior to the test-time augmentation as well?
4. Line 117 "Moreover, the diversity in architectures, training schemes, similarity measures, and tasks makes it nontrivial to combine uncertainty estimation methods with existing pretrained registration networks." - why? Aleatoric uncertainty is typically modeled using variance estimates of the output, and epistemic uncertainty imposes some prior on the model weights. Fundamentally, there is no special case in architectures or loss functions that would be incompatible with existing formulations. Consider removing this statement.
5. Line 143: "In the deep learning era, however, uncertainty estimation remains largely underexplored." - there is a wealth of work applying uncertainty estimation with deep networks (a lot is cited in the paper). Variance of a posterior is fundamentally a Bayesian concept and makes no assumptions about the model (the model can even be non-parameteric - e.g. Gaussian Processes). Related work cited in the paper are different choices of specifying appropriate priors on the model or the output space.
6. In Equation (1), $\epsilon(y)$ is typically the noise added in the problem formulation itself (to define maximum likelihood-like objectives), not in the model itself. Line 172 can be corrected to reflect this interpretation.
7. Please number all equations (Line 222, 226).
8. Line 204, "is a diffeomorphic," -> "is a diffeomorphic transform" or "is diffeomorphic,"
9. Line 322 is unclear and can benefit from further explanation
10. Fonts in all boxplots need to increase - figures are hard to read.
11. In Figure  4(a), the uncertainty maps are very qualitatively very different from each other. Which one should a practitioner interpret in their downstream task?
12. Figure 4(a), what is the uncertainty map conveying? From the map it looks like the boundaries are hardest to align - but in practice it is the easiest to align due to the high contrast of the brain with background.
13. Line 978: "To be noted, we use" -- "We note that a composition of two elastic deformations in …"
14. In Figure 10, the uncertainty map does not show the insights mentioned in the paper (i.e. if the tumor is harder to register / has more registration error because of a lack of correspondence). The paper mentions in line 459 - "Overlaying the uncertainty map on the target and warped T2 in Fig. 5a confirms that high-uncertainty regions align well with the location of the tumor" but the figure does not highlight the tumor at all. In fact, for a lot of image pairs the regions of high uncertainty do not align with the tumor at all.
15. How does Figure 10 strengthen the narrative of the paper in terms of the usefulness of the "uncertainty maps" in assessing the difficulty / potential lack of quality in registration? How does a user visualize the maps shown in Figure 10 and make a clinical decision for example?
16. Line 354: this is an inaccurate statement. Synthmorph is trained to do joint affine and deformable registration — use the “joint” parameter. The only correct portion of this statement is the fact that this reflects model’s inductive bias (i.e. lack of equivariance of the output w.r.t. moving image).
17. Line 356: "When the task matches its capacity (Fig. 3b (c)), the correlations are consistent with uniGradICON." - this meaning of this statement is unclear; consider clarifying.
18. There are several missing implementation details - how is the red line computed? Is the PCC/SC computed between per-pixel uncertainty estimates and pixelwise MSE?
19. The model performs multiple forward passes through the network to obtain different transforms. How is the computational efficiency any different from MC methods for epistemic uncertainty that also perform multiple forward passes?
20. What are the methods used in the case studies?

Overall, the paper makes statements that are either not rigorous enough, are not supported with literature, or are too bold. In my opinion, the paper can provide very valuable insights but will require significant work on improving the narration, calibrating claims, and being more rigorous with the contributions, before the quality is satisfactory for publication at a venue like ICLR.

[1] https://proceedings.mlr.press/v48/gal16.pdf

[2] Lakshminarayanan, B., Pritzel, A. and Blundell, C., 2017. Simple and scalable predictive uncertainty estimation using deep ensembles. Advances in neural information processing systems, 30.

[3] https://en.wikipedia.org/wiki/Jitter

**Questions:**

Few questions are mentioned in the weaknesses section.

---

> ### Author Response · Authors · 2025-11-20
> **Clarification of Key Conceptual Understanding (1)**
>
> We thank Reviewer 4VP3 for the thoughtful assessment and for highlighting the breadth of our experimental design across multiple architectures, anatomical regions, imaging modalities, and clinically relevant scenarios. We also appreciate the reviewer’s detailed comments and the time taken to provide comprehensive feedback.
>
> ---
>
> To ensure our responses are as clear and useful as possible, we organize them into two parts:
>
> 1. We clarify several conceptual points where our presentation may have been insufficiently clear and may have led to misinterpretations of our intended contributions. Many of the concerns raised under **Q1** fall into this category, and we address them primarily in this clarification section.
>
> 2. Given the number and granularity of the reviewer’s questions (**10 main points and 20 additional minor comments**, excluding Q1), we group them into **four thematic categories** and address each point-by-point. Throughout the response, we use Q and MQ to denote comments from the main and minor lists, respectively.
>
> ---
>
> ### **Clarification of Key Conceptual Understanding**
>
> ***C1. Reviewer’s concern: The proposed uncertainty estimation measures the equivariance error of the model, which is not a test-time augmentation scheme.***
>
> > Q1. “The interpretation of performing test-time augmentation to perform some kind of uncertainty estimation deviates from the interpretation in green box in Line 192. The latter interpretation implies that a "*true deformation*" is equivariant to linear or nonlinear diffeomorphic perturbations but the network may not be - essentially measuring the equivariance error of the model.”
>
> **1. Transformation equivariance is an expected property of the registration solver.**
>
> We apologize for not making this distinction clearer. Transformation equivariance is a property of the registration solver, not of the true deformation. For *an ideal registration solver* $f$ which always returns the true deformation, the transformation equivariance holds,
> $$f(I^A, I^B)=\\tau\\circ f(I^A\\circ\\tau, I^B).$$
> However, a learned network $f_\\theta$ generally does not satisfy this exactly.
>
>
> **2. What we measure is the variance of $\\tau\\circ f(I^A\\circ\\tau, I^B)$ with $\tau$ drawn from a distribution.**
>
> Equivariance error would require comparing the perturbed result with the unperturbed prediction:
> $$\\tau\\circ f_\\theta(I^A\\circ \\tau, I^B) - f_\\theta(I^A, I^B).$$
> Our estimator does not use $f_\\theta(I^A,I^B)$ at all.
> Instead, we compute only the variance of the “perturb–restore” predictions:
> $$\\mathrm{Var}\_{\\tau}[ \\tau\\circ f\_\\theta(I^A\\circ\\tau, I^B) ],$$
> which corresponds to Algorithm 1.
> Thus, our uncertainty reflects the stability of the predicted mapping under task-preserving perturbations, a concept that falls within the TTA scheme, rather than with equivariance error measured against the unperturbed output.
>
> **3. Practical intuition (analogy).**
>
> A helpful way to interpret our estimator is through a manual-registration analogy. Suppose a human annotator is asked to register two images using two equivalent pipelines, for example, mapping the source image to two different atlas spaces and then mapping both results to the target. If the composed transformations differ substantially across these pipelines, we would naturally interpret this variability as uncertainty in the registration task, reflecting both the ambiguity of the data and the variability introduced by the registration procedure itself. Furthermore, if multiple annotators perform these pipelines, the variability across annotators and pipelines provides a system-level notion of uncertainty for that registration task. Our method formalizes this intuition for registration neural networks.

---

> ### Author Response · Authors · 2025-11-20
> **Clarification of Key Conceptual Understanding (2)**
>
> ***C2. Reviewer’s concern: The proposed uncertainty estimation captures neither epistemic nor aleatoric uncertainty.***
>
> > Q1. “..essentially measuring the equivariance error of the model. For a (hypothetical) model that does respect the equivariance of the task and somehow produces transforms consistent with the applied transform , does it imply that the model has no estimates of uncertainty? I would argue that it is not true - as the ill-posedness of the problem does not disappear if the model becomes aware of the equivariance of the problem (and by extension the aleatoric uncertainty remains unchanged) . So the formulation is not measuring aleatoric uncertainty since the "equivariance error" can be reduced in the model by training it to respect the constraint in Equation (7) but the aleatoric uncertainty is intrinsic to the datum and cannot be reduced. On the other hand, epistemic uncertainty estimates the variance of the output (more below) under a posterior distribution over the model parameters. The proposed idea works with a pretrained network and does not impose any distribution over the model weights - so the proposed method does not compute epistemic uncertainty either. To me, this is a major fundamental issue with the paper - a simple argument showing that the proposed method captures neither epistemic or aleatoric uncertainty - and I would urge the authors to discuss and clarify this in the rebuttal.”
>
> We thank the reviewer for raising this conceptual point. We agree that our method does not fit neatly into the “epistemic vs. aleatoric” decomposition from the classic bayesian probabilistic perspective, and we clarify our intention here.
>
> **1. Equivariance is always defined with respect to a specific transform family and a specific task.**
>
> It is theoretically possible for a registration network to be perfectly equivariant to **a particular family of transforms** (e.g., translations) on a particular type of registration task and dataset. In this narrow and idealized setting, using that same transform family for test-time perturbations would produce zero variance, and our estimator with **this particular transformation family** would not reveal uncertainty.
>
> However, this does not imply that our method cannot capture task uncertainty in general. Equivariance is not universal: a model that is translation-equivariant is not by construction equivariant to nonlinear transforms.
>
> Since our framework does not restrict the choice of perturbation family, we can always probe the solver using a transform class for which equivariance does not hold. This is exactly what we demonstrate experimentally by using translations, affines, and nonlinear deformations.
>
> **2. Our method is not designed to estimate “pure epistemic” or “pure aleatoric” uncertainty.**
>
> Our method does not attempt to decompose uncertainty into the classical Bayesian categories of epistemic versus aleatoric. Instead, we share the concept used in perturbation-based and test-time augmentation (TTA) uncertainty estimation: **predictive uncertainty is measured as the variability of the output under plausible, task-preserving perturbations of the input.**
>
> Medical image registration is inherently ambiguous, and instability of the predicted deformation under task-preserving perturbations reflects exactly this ambiguity. Thus, our variance is a meaningful uncertainty estimator, even though it is not expressed within the strict Bayesian taxonomy. The high correlation between our method and the MC-dropout uncertainty empirically support this statement.
>
> Importantly, our framework also applies to probabilistic registration networks. As we show in our new experiment requested by the other reviewer, when combined with models such as TransMorph-Bayes, the measured variance naturally includes contributions from the model’s internal stochasticity in addition to perturbation-induced instability. This flexibility is central to our goal: providing a general, model-agnostic, test-time indicator of registration reliability across both deterministic and probabilistic pre-trained registration neural networks.
>
> **3. Our goal is to provide a general, model-agnostic test-time reliability indicator.**
>
> Rather than decomposing uncertainty into epistemic/aleatoric components, our method aims to provide a practical measure of registration reliability for any pretrained model. The variance reveals regions where the solver is unstable under minimal geometric changes.
>
> **4. Empirical results show a high correlation between our method and MC-dropout.**
>
> We added the comparison between our method and MC-dropout. The high correlation demonstrates that our method effectively measures uncertainty.
>
> We clarified these points in the related work section and the conclusion section in the revision.

---

> ### Author Response · Authors · 2025-11-20
> **Point-to-point Response (1)**
>
> In the following, we group the rest 30 major and minor weaknesses into four categories and address them point-by-point within each category, ordered by what we consider their importance:
> - Mathematical rigor (Q2, Q8, Q9)
> - Experimental interpretation (Q11, MQ9, MQ11, MQ12, MQ14, MQ15, MQ17, MQ18, MQ19, MQ20)
> - Statements are not rigorous, not supported by literature, or are too bold (Q4, Q5, Q6, Q7, MQ1, MQ2, MQ3, MQ4, MQ5, MQ16)
> - Writing style and typo (Q3, Q10, MQ6, MQ7, MQ8, MQ10, MQ13)
>
> ----
>
> ### **Math is not rigorous (Q2, Q8, Q9)**
> After analyzing Q2, Q8, and Q9 together, we find that the reviewer’s critique focuses on two underlying theoretical concerns:
>
> ***Q8. Additive vs. Compositional Error Modeling***
>
> We thank the reviewer for raising this thoughtful theoretical point. Registration error can be formulated compositionally,
> $\hat{\phi}^{AB} \circ (\phi^{AB})^{-1},$
> or additively,
> $\hat{\phi}^{AB} - \phi^{AB}.$ Both are mathematically valid: the compositional model reflects the geometry of diffeomorphic transformations, while an additive model reflects the standard regression view of predicting continuous-valued deformation fields. Our goal is to undertand how the variance relates to the error, i.e., to rewrite
> $$\\mathrm{Var}= g(\\hat{\\phi}^{AB} \\circ (\\phi^{AB})^{-1})$$
> or
> $$\mathrm{Var}= g(\hat{\phi}^{AB} - \phi^{AB})$$
> so that we can explain why variance and error correlate and how to interpret variance in practice.
>
> In practice, however, registration error is almost always computed additively (e.g., Euclidean distance between warped and target landmarks, which corresponds to $\hat{\phi}^{AB}(\cdot) - \phi^{AB}(\cdot)$. **To keep the theoretical analysis consistent with empirical evaluation, we therefore adopt the additive error model**.
>
> Furthermore, our uncertainty estimator is designed to be model-agnostic and does not assume that predicted transformations are diffeomorphic. In realistic settings, networks may not enforce diffeomorphism, and even diffeomorphic architectures can yield non-diffeomorphic outputs under poor training (e.g., early stop during training or inappropriate hyperparameters). A compositional error model would apply only to a restricted set of ideal settings, whereas the additive model holds universally.
> For these reasons, we use the additive error model: it is the most general, empirically consistent, and practically interpretable formulation for our analysis.
>
>
>
> ***Q2,Q9. Theoretical Relation to Epistemic/Aleatoric Uncertainty***
>
> We appreciate the reviewer’s concern about the connection to standard epistemic and aleatoric definitions. As we discussed in the clarification, our method is not intended as a decomposition of these two components, nor do we present it as such. Instead, we measure predictive uncertainty through the variability of the output under task-preserving perturbations.
>
> This does not imply a lack of rigor. Rather, it reflects a different uncertainty formulation. Importantly, we do not claim that the terms in our decomposition correspond to epistemic or aleatoric quantities. As derived in Sec. 3.3, the decomposition explains how perturbation-induced variance relates to registration error, clarifying when it reflects error and when it may not.
>
> Finally, we emphasize that our framework is compatible with both deterministic and probabilistic registration networks; when applied to models with internal stochasticity, the measured variance naturally captures both model-intrinsic and perturbation-induced variability.
>
> In summary, our goal is to provide a model-agnostic uncertainty indicator, not a Bayesian epistemic/aleatoric decomposition. We clarified this in the conclusion section of the revised manuscript.
>
> -----
>
> ### **Experiments interpretation (Q11, MQ9, MQ11, MQ12, MQ14, MQ15, MQ17, MQ18, MQ19, MQ20)**
> In this section, we address the concerns and questions about the experimental setting and the interpretation of the experiment results.
>
> ***Q11. Interpretation of Fig. 3 correlations***
>
> The reviewer states that Fig. 3 does not show meaningful trends between PCC/SC and registration performance.
>
> As described in the main text and confirmed by other reviewers, the trends are consistent and interpretable:
> - Strong correlation when the ground-truth deformation is translation or affine (Fig. 3a(a–b), Fig. 3a(e–f)).
> - Moderate or weaker correlation when the true deformation is nonlinear (Fig. 3a(c–d), Fig. 3a(g–h)).
> - This trend appears both in brain MRI (Fig. 3a) and abdomen CT (Fig. 3a(e–h)).
>
> When replacing uniGradICON with SynthMorph (Fig. 3b), the correlation in the linear GT cases decreases significantly. This is expected because SynthMorph is only trained on nonlinear (SVF) transformations, so translation/affine perturbations fall outside its inductive bias. This explains the drop in correlation in Fig. 3b(a–b).

---

> ### Author Response · Authors · 2025-11-20
> **Point-to-point Response (2)**
>
> ***MQ9. Line 322 clarification***
>
> Line 322 intends to explain why correlation drops for nonlinear ground-truth deformations. Two factors contribute:
> - Nonlinear ground truths induce localized errors.
>
>     When the underlying deformation is nonlinear, A well-trained nonlinear model rarely produces a global error (e.g., a uniform translation shift). Errors tend to be spatially localized (see Fig. 4a(R3)).
>
> - Linear perturbations are easy for uniGradICON.
>
>     For nonlinear GT tasks, uniGradICON consistently finds the same correspondence under linear perturbations, producing low variance but nonzero registration error, leading to diminished correlation (Supp. C.3 scenario (i-b)).
>
> ***MQ11. Which uncertainty map should a practitioner use?***
>
> In Fig. 4(a), the three linear perturbations (translation/scale/shear) yield very similar uncertainty maps, while the nonlinear perturbation produces a different one.
>
> Our recommendation for practical use is to compute uncertainty using:
> - one linear perturbation, and
> - one nonlinear perturbation,
> to capture complementary behaviors.
>
> ***MQ12. Why do boundary regions show high uncertainty in Fig. 4(a)? They should be easy to register.***
>
> The uncertainty map in Fig. 4(a) does not imply that brain boundaries are difficult to align in general. In this particular slice:
> - The boundary region might have lower contrast and correspondingly higher registration error, so the uncertainty correctly reflects this difficulty.
> - The heatmap is also relative within the slice, so a region may look high simply because the rest of the regions have low uncertainty in the visualized slice.
>
> Crucially, this is not a consistent pattern: in Fig. 4(a) (third row), Fig. 5, and Fig. 6(a), the brain boundary does not show elevated uncertainty. Thus, the boundary is not systematically highlighted unless the local alignment is genuinely less reliable.
>
>
> ***MQ14. Fig. 10 does not always highlight the tumor***
>
> In Fig. 10, panels (a, b, c, e) show uncertainty concentrated around the tumor region, and panel (d) does not highlight the tumor because both source and target contain similar abnormalities, reducing correspondence ambiguity. Here, “align” refers to co-localization of high-uncertainty regions and abnormal structures, not perfect geometric alignment.
>
> ***MQ15. How Fig. 10 supports the narrative***
>
> Fig. 10 illustrates that uncertainty rises when the structures in the two images do not correspond. In large-scale studies, uncertainty maps allow users to:perform automated QC, detect abnormal image pairs, and exclude problematic cases from downstream analysis.
>
>
> ***MQ 17. Meaning of “when the task matches its capacity”***
>
> We clarify that this refers to SynthMorph’s inductive bias:
> - We use the nonlinear model from SynthMorph as we explained in Sec. 5 Backbones.
> - For nonlinear GT tasks (Fig. 3b(c)), its correlation pattern matches uniGradICON (Fig. 3a(c)).
> - For tasks outside its capacity (large translations/affine), correlation drops.
>
>
> ***MQ18. How the red line and correlations are computed***
>
> The red line represents the mean registration error, computed as voxel-wise Euclidean (L2) norm between the predicted and ground-truth transformed positions $\| \phi^{AB}(y) - \hat{\phi}^{AB}(y) \|_2 $ with $y\in\Omega^B$, averaged over the ROI and across the dataset. The Pearson and Spearman correlations are computed between the uncertainty map and the voxel-wise registration error. These correlations are then averaged within each ROI and across the dataset to obtain instance-level (Fig. 4 (b)) and dataset-level (Fig. 3) correlation scores. We add these details in the Supp C.2
>
> ***MQ19. Computational efficiency vs. MC methods***
>
> We agree. Our method requires multiple forward passes through the network, as do Monte Carlo dropout or ensembles. Therefore, its computational cost is comparable to standard MC-based epistemic uncertainty methods.
>
>
> ***MQ20. Methods used in case studies***
>
> All case studies use uniGradICON as the registration backbone. We will ensure this detail is stated clearly in Supplementary C.5.

---

> ### Author Response · Authors · 2025-11-20
> **Point-to-point Response (3)**
>
> ### **Statements are not rigorous, not supported by literature, or are too bold ( Q4, Q5, Q6, Q7, MQ1, MQ2, MQ3, MQ4, MQ5, MQ16)**
> We add necessary citations, improving the language clarification regarding Q6, MQ1, MQ2, MQ5, MQ16 in the revised manuscript.
>
> ***Q4. Clarification of computational expense statement***
>
> We appreciate the reviewer’s discussion, but our statement was referring specifically to classical probabilistic registration methods, which rely on iterative optimization or sampling over deformation parameters and therefore do not scale well. The sentence was meant to contrast these traditional approaches with learning-based probablistic methods that follow in the text.
>
> ***Q5. Claim of novelty in Line 88***
>
> The novelty we intend to communicate is that, to our knowledge, ours is the first work to
> 1. formalize perturbation-based uncertainty specifically for image registration based on its transformation-equivariant structure, and
> 2. provide a theoretical decomposition linking perturbation-induced variance to registration error.
>
> To avoid misunderstanding, we revised the sentence to:
> >“We present the first theoretical analysis of the test-time perturbation-based uncertainty in image registration, decomposing the perturbation-induced variance with respect to registration error and revealing the relationship between the two.”
>
> In addition, we have explicitly discussed the connection between our method and TTA-based uncertainty in the related section.
>
> ***Q7. Distinction between test-time augmentation and Bayesian uncertainty***
>
> We agree that TTA-based uncertainty reflects properties more closely aligned with aleatoric uncertainty, whereas Bayesian formulations explicitly model epistemic uncertainty. To better situate our work in the broader landscape, we revised the related-work section to explicitly differentiate between Bayesian epistemic-uncertainty methods and test-time perturbation methods, clarifying which type of uncertainty each class of approaches targets.
>
>
> ***MQ3. Lack of discussion of classical registration and whether TTA applies***
>
> Our paper focuses on uncertainty estimation for pretrained learning-based models. Thus, we cite several representative classical methods such as ANTs, elastix, NiftyReg, and Deeds but do not intend to discuss in detials. Our uncertainty estimation presumably could be applied to classical/iterative methods, and we note this explicitly as a potential direction for future work.
>
>
> ***MQ4. Statement that combining existing uncertainty methods with pretrained networks is “nontrivial”***
>
> We agree that, in principle, Bayesian and ensemble-based uncertainty methods can be applied to many architectures. Our point was about practical integration, not theoretical incompatibility. In modern registration networks, introducing variational priors or ensembles is often nontrivial due to
> 1. architectural complexity (e.g., multiresolution pipelines where latent variables would need to be propagated across stages),
> 2. sensitivity to additional Bayesian/variational hyperparameters, and
> 3. the need to retrain large registration models from scratch, often requiring weeks of computation for models such as uniGradICON or SynthMorph.
>
> In contrast, our test-time perturbation approach is model-agnostic, requires no architectural changes, and works directly with any pretrained network. We have revised the manuscript to make clear that our point concerns practical integration challenges, not conceptual limitations of Bayesian frameworks.
>
> ---
>
> ### **Writing style and typo (Q3, Q10, MQ6, MQ7, MQ8, MQ10, MQ13)**
> We addressed Q3, MQ7, MQ8, MQ10, and MQ13 in the revised manuscript.
>
> ***Q10. Naming of “bias jitter” term***
>
> The reviewer indicates that “bias jitter” may be misleading for the first term and that the second term in Eq. (11) is not a bias but a covariance term. We want to clarify that we decompose the variance w.r.t. the registration error in Eq.(11):
> - The first term is the mean of the variance of the error. We name this intrinsic spread because it reflects the intrinsic variability of the error across perturbations.
> - The second term is the variance of the mean of the error. We refer to this as bias jitter, since it measures the variability (“jitter”) of the error’s mean (i.e., its bias) across perturbations.
>
>
> ***MQ6. Interpretation of ε in Equation (1)***
>
> We thank the reviewer for the suggestion. In our work, $\\epsilon$ is explicitly defined as the registration error term, not a noise term.
> Since the notation is clearly defined and used consistently throughout the paper, we retain $\\epsilon$ to avoid introducing additional symbols.
>
>
> ***MQ10. Font size in boxplots***
>
> Due to the space limitation, it is very difficult to increase the font size, considering the figures are already extremely compact. We added the original high-resolution PNG files in a zip file and submit them as the supplementary material.

---

> ### Author Response · Authors · 2025-11-24
> **Follow-up on the Discussion**
>
> Dear Reviewer 4VP3,
>
> We greatly appreciate the efforts you have made to improve the quality of our manuscript. It will also be appreciated if you have the chance to look at our detailed answers to your questions and let us know your thoughts. We are super happy to provide further details and clarifications if you still have any questions.
>
> Thanks for your time.
>
> Best,
>
> Authors of paper #6128

---

### Author Response · Authors · 2025-11-20
**General Comment to All Reviewers**

We sincerely thank the reviewers for their thoughtful evaluations and constructive feedback. We appreciate the **recognition of the practical value of a general, test-time uncertainty estimator** that can be applied to **any pretrained registration network** without architectural modifications or retraining (Reviewers Edea, 1SzL, 3ar4), an ability noted as important for clinical pipelines (Reviewer 4VP3). Reviewers Edea, 1SzL, 4VP3 also **highlight the breadth of our experimental evaluation**, spanning four anatomical regions, multiple perturbation families, and two representative registration backbones.

In response to reviewers' suggestions,
- We have added **a new experiment comparing our method with MC-dropout–based uncertainty** using the publicly available probabilistic registration model TransMorph-Bayes. The experiment shows a high correlation between our method and MC-dropout.
- We have revised the manuscript (highlighted in red) to improve clarity and presentation throughout.

We greatly appreciate the reviewers’ insights, and we believe the revisions have strengthened both the clarity and completeness of the paper.

**As a result, the contributions of the latest revision are as follows**:
- We introduce a test-time uncertainty estimation framework that can be applied to any registration network without retraining or model modification.
- We present the first theoretical analysis of the test-time perturbation-based uncertainty in image registration, decomposing the perturbation-induced variance with respect to registration error and revealing their relationship.
- We conduct extensive experiments across four anatomical structures (brain, cardiac, abdominal, lung) and three registration backbones to evaluate (i) the correlation between the estimated uncertainty maps and the true registration error, (ii) comparisons between the proposed method and one existing uncertainty estimation approach, and (iii) two case studies with real transformations and anatomically inconsistent image pairs.

---

### Author Response · Authors · 2025-12-02
**Discussion Summary to AC**

Dear Area Chair,

Our work has received three positive reviews [$\text{\textcolor{blue}{3ar4}}$, $\text{\textcolor{blue}{1SzL}}$, $\text{\textcolor{blue}{Edea}}$] and substantially more critical review from Reviewer $\text{\textcolor{blue}{4VP3}}$, whose evaluation, we believe, is partly based on misunderstandings, internal inconsistencies, overly harsh and not fully supported by the issues raised.

Reviewers [$\text{\textcolor{blue}{3ar4}}$, $\text{\textcolor{blue}{1SzL}}$, $\text{\textcolor{blue}{Edea}}$] agree that,
- We provide a valuable approach to an important problem for clinical pipelines.
- Our approach is general and practical to estimate the uncertainty for any pre-trained registration networks without architectural modifications or retraining.
- Our empirical results are promising and encouraging.

In addition, Reviewers [$\text{\textcolor{blue}{Edea}}$, $\text{\textcolor{blue}{1SzL}}$, $\text{\textcolor{blue}{4VP3}}$] highlight the breadth of our experimental evaluation, spanning four anatomical regions, multiple perturbation families, and two representative registration backbone networks.

In response to reviewers' suggestions,
- We have added **a new experiment comparing our method with MC-dropout–based uncertainty** using the publicly available probabilistic registration model TransMorph-Bayes. The experiment shows a high correlation between our method and MC-dropout.
- We have added one paragraph to discuss the rationale behind the choice of the additive error model.
- We have revised the manuscript (highlighted in red) to improve clarity and presentation throughout.

We appreciated all the constructive suggestions and discussions that helped us to further strengthen our work. We believe that we have fully addressed the concerns and questions of Reviewers [$\text{\textcolor{blue}{3ar4}}$, $\text{\textcolor{blue}{1SzL}}$, $\text{\textcolor{blue}{Edea}}$]. We have also addressed, to the best of our ability, the actionable comments and concerns raised by Reviewer$\text{\textcolor{blue}{4VP3}}$ in both the revision and the response.

----

We want to bring to your attention that Reviewer $\text{\textcolor{blue}{4VP3}}$'s comments appear disproportionately negative relative to the evidence in the review.

Reviewer $\text{\textcolor{blue}{4VP3}}$ summarizes the review as follows and suggests strong rejection.
> In my opinion, the paper can provide very valuable insights, but will require significant work on improving the narration, calibrating claims, and being more rigorous with the contributions, before the quality is satisfactory for publication at a venue like ICLR.

The severity of these problems is exaggerated. As we detail in the point-to-point response, many of the issues cited here arise from misreadings or are straightforwardly addressable by clarifying the writing or adding references. Specifically, several concerns reflect misunderstandings of our intent (Q4, Q5, Q7, MQ4), others are not directly related to our main contribution (MQ3), and the remaining points (Q6, MQ1, MQ2, MQ5, MQ16) are addressed in the revision through improved exposition and additional citations. None of these points invalidate our core contributions: the proposed test-time, model-agnostic uncertainty estimation framework for pre-trained registration networks, the accompanying theoretical analysis of the relationship between the uncertainty and errors, or the experimental results. For this reason, we feel that a strong-reject recommendation based on these issues is disproportionately severe.

The most critical concern raised by $\text{\textcolor{blue}{4VP3}}$ is that our method does not measure uncertainty at all (Q1). However, this appears inconsistent with Reviewer $\text{\textcolor{blue}{4VP3}}$’s own comments in Q5, where the Reviewer emphasizes that test-time augmentation–based uncertainty has been studied in the literature and that our approach falls into this general scheme, arguing instead that our novelty claim (“the first theoretical account of perturbation-based uncertainty in image registration…”) should be moderated. This tension indicates that the Q1 statement may overstate the issue: in Q5, the reviewer explicitly acknowledges that our method performs test-time uncertainty estimation, yet in Q1 argues that it does not measure uncertainty at all. These two positions are inconsistent.

Given the overly harsh nature of $\text{\textcolor{blue}{4VP3}}$'s evaluation and the mixed opinions across the review, we try our best to respond to $\text{\textcolor{blue}{4VP3}}$'s comments in detail. **We respectfully ask that you consider these clarifications and our revisions when making the final decision.**

Best,

Authors of paper \#6128

---

### Meta-Review · Area_Chair_CWRt · 2026-01-11

**Summary:**

Reviewer 4VP3 (score 0)
had major concerns about the correctness of the claims about quantifying uncertainty, mathematical rigor in both the definitions/claims and the exposition of the claims, and the experimental claims that estimates of the uncertainty correlate with the performance of registration. The reviewer also suggested a number of ideas on improving the definitions and the narrative to make it tighter.

Reviewer 3ar4 (score 6)
This is a short review and scant on technical comments.

Reviewer 1SzL (score 6)
commented that the proposed metric does not seem to capture epistemic uncertainty and that the theoretical development is informal. They also made similar comments as those of the first reviewer regarding additive prediction error.

Reviewer Edea (score 8)
commented on inadequate comparisons to existing work on uncertainty estimation.

This paper resulted in a robust set of reviews and an elaborate discussion. The main point (raised by multiple reviewers) was that it is unclear how this perturbation based quantity relates to uncertainty and what exactly it measures. This point went largely unaddressed in the rebuttal. The central thesis of the author rebuttal was that although their quantity does not relate to Bayesian uncertainty per se, it is certainly useful. While this is indeed correct, it is important to write the paper in such a way that the point comes across, e.g., the title of the paper and all the paper talks about developing a new approach for uncertainty estimation. It would help the manuscript to tighten this narrative and clearly define what uncertainty is being measured. There were also some comments on the mathematical exposition of the paper, that can be improved in the final version.

To add my two cents: I believe the intrinsic spread and bias jitter that is being calculated relate to what in learning theory one measures as “stability”, e.g., how similar are two solutions returned by SGD upon changes in the dataset/model class/hyper-parameters. The case in point measures sensitivity of the registration solution to changes in the model class, whether the architecture correctly captures the equivariance implied by the choice of augmentation at test time. It would be much more appropriate to call this quantity sensitivity to test-time augmentations---and this takes nothing away from its usefulness. If one does so however, it is important to study what kinds of sensitivity is necessary and expected, what downstream research questions can be addressed by measuring the sensitivity, etc.

**Reviewer Concerns:**

Reviewer 4VP3
This was a very long and detailed review. The authors have provided a long and elaborate rebuttal. To summarize, they agree that the quantity computed in this paper (variance of the transformed warp field on images augmented using the transformations) is not really an estimate of the uncertainty. This does not mean it is not useful...but does mean that the narrative needs to change. I would have expected the authors to modify the narrative in the paper significantly in light of this conversation. This is the most important point of the review/response. Next, there are a number of clarifications provided by the authors regarding the empirical results. These are somewhat inadequate. The authors have proposed changes to the narrative that clarify some of the concerns raised in the review (claims of novelty, citations to existing literature, discussion between TTA and Bayesian uncertainty).

Reviewer 3ar4
The authors have provided a good response to the comments made by the Reviewer (about whether image differencing would be comparable, and some intuition behind figures in the paper).

Reviewer 1SzL
The rebuttal made similar points as those of the rebuttal to the first reviewer, namely that the uncertainty metric does not cleanly connect to Bayesian uncertainty but can correlate to the residual. The authors also argued for the fact that although this idea does not seem very new by itself, it has not been explored in the literature yet.

Reviewer Edea
The authors provided some new experiments comparing their uncertainty maps to Monte Carlo-like uncertainty estimated using dropout.

**Reviewer Scores:**

Reviewer 4VP3
In spite of the long rebuttal, I think the main points raised in the review are not addressed. Perhaps they cannot be, without a full rewrite of the paper. I suspect that the score would not have improved much, e.g., it would still lean towards a reject.

Reviewer 3ar4
I suspect that this score is likely to reduce based on the comments by other reviewers. At best it might stay at 6. This is a short review and scant on technical comments. I will discount it in my recommendation.

Reviewer 1SzL
I suspect that this score is likely to reduce based on the comments by other reviewers. At best it might stay at 6.

Reviewer Edea
I suspect that this score is likely to reduce based on the comments by other reviewers.

---

### Decision · Program_Chairs · 2026-01-26

Reject